# Sparse Labels Node Classification: Unsupervised Learning for Mentoring Supervised Learning in Sparse Label Settings

## Abstract

Despite their huge success, Graph Neural Networks (GNNs) still require lots of labeled examples (per class) at training time in order to perform well on the Semi-Supervised Node Classification (SSNC) task. This is a major drawback since labels are usually expensive and time-consuming to get. Though several attempts have been made to address this problem, most attempts still require; a significant amount of labeled examples for at least some classes (considered base classes), as well a minimum amount of labels per class (for other classes). In this work, we attempt to alleviate these hard requirements. Our problem thus differs from the traditional SSNC settings in the sense that in this work we try to address the setting in which we only have extremely few labeled nodes seen at training time, and in addition, these labeled nodes are not provided (chosen) on a per-class basis. We name this task Sparse Labels Node Classification (SLNC). To address this problem, we Estimate Label Information (ELI) from a pseudo space by leveraging unsupervised learning techniques. We use this estimated label information to enhance reformulations of well-known semi-supervised learning (SSL) frameworks, as well as guide the labeled nodes selection process for training. We show that our approach outperforms baselines on SLNC by 10-20% when the number of labeled nodes seen at training is extremely few.

## 1 Introduction

Classification which often serves as a preliminary (Serengil & Ozpinar, 2021; Redmon et al., 2016) or final step (He et al., 2016; Kipf & Welling, 2017; Li et al., 2019) for several applications is a ubiquitous task in several domains ranging from images (Li et al., 2019), to shapes (Sun et al., 2018), to graphs (Di et al., 2020; Xu et al., 2019), and nodes (Velickovic et al., 2018; Delalleau et al., 2005; Kipf & Welling, 2017). Though our work may be generalized to several if not all of these domains, that is a monumental task requiring a lot of work and further analysis. So, in this work we will restrict ourselves to graphs, particularly to node classification.

Given an attributed graph $\mathcal{G}_A = (\mathcal{V}, \mathcal{E}, X)$, where $\mathcal{V} = \{v_1, \cdots, v_n\}$ is the set of nodes and $\mathcal{E}$ is the set of edges in $\mathcal{G}_A$. $\mathcal{E}$ can be represented by an adjacency matrix $A = \{0, 1\}^{n \times n}$. The set of nodes is partitioned into a set of labeled nodes $\mathcal{V}_L$ and a set of unlabelled nodes $\mathcal{V}_U$. The node attributes are represented by the matrix $\boldsymbol{X} = [\boldsymbol{x}_1; \cdots; \boldsymbol{x}_n] \in \mathbb{R}_+^{n \times d}$, where $\boldsymbol{x}_i \in \mathbb{R}_+^d$ is a non-negative attribute vector of node $v_i$. The aim of semi-supervised node classification (SSNC) is to learn a function $\mathcal{F}_\Theta(\boldsymbol{A}, \boldsymbol{X}) : \mathcal{V} \to \mathcal{Y}$ that maps the nodes in the graph to their corresponding labels, with the learning process being guided by minimizing (or maximizing) a suitable loss function $\min_\Theta \mathcal{Z}(\mathcal{F}_\Theta(\boldsymbol{A}, \boldsymbol{X}), \mathcal{Y}_{\mathcal{V}_L})$ where $\mathcal{Y}$ is the label set and $|C| = c$ is the number of unique node labels (referred to as classes $C$), and $\mathcal{Y}_{\mathcal{V}_L}$ is the set of labels of labeled nodes $\mathcal{V}_L$ used for training.

SSNC's performance depends on (a) the design of $\mathcal{F}_\Theta(\boldsymbol{A}, \boldsymbol{X})$, (b) the design of the loss $\mathcal{Z}(\cdot)$, as well as (c) the optimization technique employed.

For SSNC, existing baselines in order to perform well are often provided with a substantial amount of labeled nodes at training which are usually provided on a per-class basis. However, in real life (1) it is difficult to obtain such amount of labels, and (2) it is even more difficult to do so on a per-class base. In this light, we propose a setting where these two requirements are relaxed i.e., we propose the setting where (1) we have access to only very few labels in general and (2) these labels are

not necessarily supplied on a per-class basis but randomly (i.e., some classes may not even have labels at training time). We term this setting the Sparse Labels Node Classification Problem. The only requirement we maintain however is that the number of unique classes $c$ is known in advance. Moreover, we propose a framework ELI (Estimating Label Information) suitable for SLNC, which when incorporated into the existing SSNC baselines yields 10-20% better performance for SLNC.

One may group our main contributions as follows:

- We introduce the Sparse Labels Node Classification (SLNC) problem,
- for SLNC, we propose a suitable framework built around estimating label information (ELI),
- we show that with ELI, the performance of existing SNNC baselines increases by 10-20% for SLNC. We equally show in Appendix D that even in the case where very few nodes are used on a per-class basis, ELI-enhanced baselines still outperform other baselines by 10-20%.

To these ends, we organize the sections in the rest of the paper as (a) related work, (b) sparse labels node classification (SLNC), (c) ELI, a framework for sparse label classification, and generalization of the framework, (d) experiments, (e) limitations, and (f) conclusion.

Table 1: Notations

| Name | Description | Name | Description |
|---|---|---|---|
| $\mathcal{G}_A$ | The original graph | $\boldsymbol{A}$ | The original graph adjacency |
| $\mathcal{G}_H$ | The pseudo label graph | $\boldsymbol{A}_{\mathcal{G}_H}$ | The pseudo label adjacency |
| $\mathcal{G}_Y$ | The ground truth label graph | $\boldsymbol{A}_{\mathcal{G}_Y}$ | The ground truth label graph adjacency |
| $\mathcal{V}$ | A set of nodes | $v_i$ | The $i$-th node ($i = 1, 2, \cdots, n$) |
| $\boldsymbol{X}$ | An attribute matrix of all nodes | $\mathbf{x}_i$ | The node attribute vector of $v_i$ |
| $\mathcal{E}$ | A set of edges | $e_j$ | The $j$-th edge ($j = 1, 2, \cdots, m$) |
| $\mathcal{V}_L$ | A set of labeled nodes | $\mathcal{V}_U$ | A set of unlabeled nodes |
| $\boldsymbol{D}$ | A diagonal matrix of node degrees | $\boldsymbol{L}_{sym}$ | The normalized symmetric Laplacian |
| $\boldsymbol{Y}$ | The ground truth Label matrix | $\boldsymbol{Y}_{i,:}$ | The label indicator for node $i$ |
| $|C| = c$ | The number of unique node labels | $l_c$ | Unique node pseudo classes |
| $\boldsymbol{H}$ | Pseudo classes assignment matrix | $\boldsymbol{H}_{i,:}$ | pseudo label of node $i$ |
| $l_H$ | Number of nodes with smallest loss in pseudo space | $l_R$ | Number of randomly chosen nodes |
| $\mathcal{N}(\boldsymbol{L}_{sym})$ | The graph filter function | $\boldsymbol{F}$ | Denoised node attributes $\boldsymbol{F} = \mathcal{N}(\boldsymbol{L}_{sym})\boldsymbol{X}$ |
| $\mathcal{Z}(\cdot)$ | Loss function | $\mathcal{F}_\Theta$ | Neural network |
| $\boldsymbol{R}$ | predicted labels | $\boldsymbol{A}_A$ | Averaged graph adjacency |

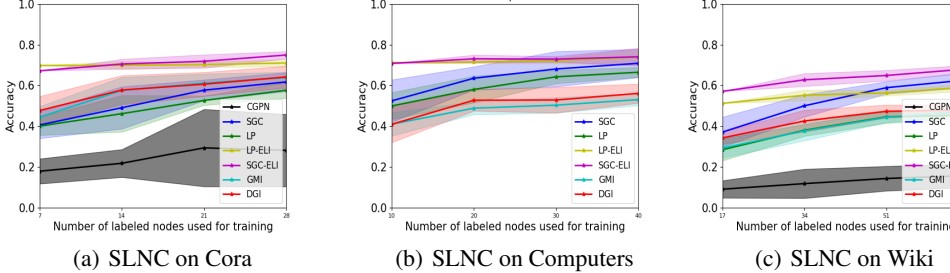

(a) SLNC on Cora     (b) SLNC on Computers     (c) SLNC on Wiki

Figure 1: Illustration of the Sparse Labels Node Classification (SLNC) accuracy of baselines using our estimation of label information (ELI) vs baselines not using ELI (see Section 3). Each experiment is run 10 times (see parameter settings in Section 5.5). All baselines with ELI achieve up to 10-20% more classification accuracy compared to those without ELI. Darker lines are the mean for each plot and lighter colors are the standard deviations .

## 2 RELATED WORK

For learning on graphs, both in the supervised, semi-supervised, and unsupervised learning communities, it has generally been agreed upon that the neighborhood of a node is of utmost importance (Delalleau et al., 2005; Velickovic et al., 2018; Kipf & Welling, 2017; Hamilton et al., 2017; Fey et al., 2020; Wang et al., 2019; Huang et al., 2019), whether for homophilic graphs (neighbors tending to share the same labels) or for heterophilic graphs (neighbors tending to share different labels). This observation has led to designing $\mathcal{F}_\Theta(\cdot)$ and $\mathcal{Z}(\cdot)$ in such ways as to leverage this information

contained in the neighborhood connectivity of a node by utilizing the famous graph convolution (leading to graph convolution neural networks GCNs (Kipf & Welling, 2017; Wu et al., 2019; Zhang et al., 2019b)), or approximation of the convolution (Hamilton et al., 2017), or attention strategies (leading to graph attention GAT (Velickovic et al., 2018)), or other neighborhood aggregation strategies (Di et al., 2020; Gasteiger et al., 2019; Chien et al., 2021; Xu et al., 2019) tailored to address homophily, heterophily, and other challenges. For $\mathcal{Z}(\cdot)$, to enforce neighborhood statistics, the Laplacian regularization strategy has been broadly adopted (Kipf & Welling, 2017; Delalleau et al., 2005). However, a major drawback of these SSNC baselines is the requirement of a substantial amount of labels needed (generally on a per-class basis) at training time to learn a good $\mathcal{F}_\Theta(\cdot)$ (Ding et al., 2020; Wang et al., 2022). This is a hard requirement since labels are usually expensive and time-consuming to get, and in scenarios where the graphs are arbitrarily large, getting a consistent amount of labels becomes prohibitive. Moreover, in real life, it is extremely difficult to select labels on a per-class basis. Figure 1 Illustrates this problem faced by existing SSNC baselines by showing that LP (Delalleau et al., 2005), and SGC (Wu et al., 2019) perform poorly on various graphs such as those of Cora Gasteiger et al. (2019), Computers Chen et al. (2022) and Wiki Zhang et al. (2019b) when using extremely few random labels. In this setting, their performance is about 10-20% below their performance when using sufficient labels (see Appendix D, and their original papers).

Pre-training approaches (Veličković et al., 2019; Peng et al., 2020; Ma et al., 2021) were proposed to enhance the performance of SSNC baselines by learning features in an unsupervised manner and using these learned features in a semi-supervised manner for different downstream tasks (SSNC in this case). Though such pre-training helps to improve the performances of GNN when training on the target graph, these methods still require a substantial amount of labels per class when training on the downstream task to perform well. This is shown in figure 1 by the poor performance of the pre-training baselines DGI (Veličković et al., 2019) and GMI Peng et al. (2020) when only very few labels are used for the downstream task.

There has equally been an attempt to address the very sparse label setting by Wan et al. (2021), however, their model requires the labeled nodes to be selected on a per-class basis. As such it does not perform well in the SLNC setting as shown in figures 1(a) and 1(c). Moreover, it takes a lot of time, so we did not use it for larger datasets.

Other methods (Lan et al., 2020; Ding et al., 2020; Wang et al., 2022; Liu et al., 2021; 2019; Zhang et al., 2019a) have tried to address the few-shot scenario. In this setting, at training time some classes termed base classes have abundant nodes seen (usually per class), while other classes termed novel classes only have a few nodes seen (these few nodes from the novel classes are called the support set), and evaluation/testing is done on unlabelled nodes (called the query set) from the novel classes. Baselines in this category usually employ meta-learning on the labeled nodes from the base classes, generalizing to the support nodes from the novel classes, and testing on query nodes from the novel classes. Since these methods need a substantial amount of labels (for the base classes used for the meta-learning phase), they are not suitable for SLNC because in SLNC only very few labels are available for training for all classes, and maybe some classes may not even have labels (since labels are chosen at random and not on a per-class basis).

Another related line of work is that of robustness and fairness under domain shift (Rezaei et al., 2021; Liu & Ziebart, 2014; Yang et al., 2020; Guidotti et al., 2018; Celis et al., 2019; Zhai et al., 2021), as indeed having very few labels may often result in a scenario where the distribution of labels in the training set is quite different from the distribution of labels in the test set Wang et al. (2022). In this domain, several $\mathcal{Z}(\cdot)$ (Liu & Ziebart, 2014; Chen et al., 2022), as well as optimization strategies (Rezaei et al., 2021), have been proposed to address the challenge of distribution change. To our disadvantage, in this work, we do not include comparisons with these approaches due to lack of time (we would joyfully do so in extensions to this work).

Other works in other domains (such as Vision) that are related to our work include those of Neighbourhood Constraint Regularization (Iscen et al., 2022), where a neighborhood regularizer is used to enforce neighborhood preservation when labels are noisy, Co-Teaching (Han et al., 2018) where two networks are trained in parallel and each networks prediction are used to refine and enhance those of the other, Mentoring (Jiang et al., 2018) where a network is trained and its predictions are used to train a subsequent network, and others (Malach & Shalev-Shwartz, 2017; Reed et al., 2015).

## 3 SPARSE LABELS NODE CLASSIFICATION

In this section, we will formally introduce sparse labels node classification.

**Definition 3.1.** *Sparse labels node classification is the semi-supervised node classification problem* $\min_\Theta \mathcal{Z}(\mathcal{F}_\Theta(\boldsymbol{A}, \boldsymbol{X}), \mathcal{Y}_{\mathcal{V}_l})$, *aiming at minimizing the loss* $\mathcal{Z}(\cdot)$ *over the parameters* $\Theta$, *in a scenario where there are only very few numbers of labeled nodes* $\mathcal{V}_L$, *and these labeled nodes are not chosen on a per-class basis, but randomly over the entire set of nodes* $\mathcal{V}$ *(as will be the case in real life) and the rest of the nodes in the graph are used for testing.*

In this work, for fair evaluation purposes we assume that the number of classes $c$ is known in advance, and we choose the number of labeled nodes $|\mathcal{V}_L| = l$ as a multiple of the number of unique classes $c$ present in the graph i.e., the labeled nodes $\mathcal{V}_L$ are not chosen per class, but the total number of labeled nodes $l$ for training is determined as a multiple of the number of classes $c$ and then the nodes $\mathcal{V}_L$ are randomly chosen over the entire node set $\mathcal{V}$.

## 4 ELI (A PROPOSED FRAMEWORK) FOR SPARSE LABELS NODE CLASSIFICATION)

In this section, we propose to estimate label information (ELI) for SLNC. The intuition behind ELI is the fact that the neighborhood distribution and attributes of the nodes are not sufficient to be able to achieve good performance in the presence of sparse labels. This is depicted in figure 1 by the poor performance of neighborhood-based methods such as Label Propagation (Delalleau et al., 2005) and graph convolutional neural networks (Wu et al., 2019), as well as the poor performance of feature learning (pre-trained) methods such as DGI (Veličković et al., 2019) and GMI (Peng et al., 2020) for the SLNC task.

Our intuition is that to perform well in such a setting, (a) randomly selected nodes whether on a per-class basis or randomly over all classes may not be representative of the classes in the graph, and (b) The message-passing framework of GNNs may not be well-adapted given that the label distribution over the entire graph may not be well captured by the adjacency matrix of the graph. As such, one needs some amount of information about the label distribution of the nodes over the entire graph, as well as a strategy for choosing a representative set of nodes as labeled nodes $\mathcal{V}_L$.

We thus propose ELI, a framework comprising four steps, namely: (1) label distribution estimation, (2) key nodes selection as labeled nodes, (3) label distribution incorporation in sparse labels classification, (4) optimize, and (5) generalize the proposed framework.

### 4.1 LABEL DISTRIBUTION ESTIMATION

In this step, we attempt to estimate the label distribution of nodes on a graph as a function of the node feature similarity and the graph structure *in an unsupervised way*. Formally, given the graph structure of an attributed graph $\mathcal{G}_A$ represented as an adjacency matrix $\boldsymbol{A} \in \mathbb{R}^{n \times n}$, where $n = |\mathcal{V}|$ is the number of nodes in the graph, and the node features (attributes) $\boldsymbol{X} \in \mathbb{R}^{n \times d}$, where $d$ are the number of attributes per node, our goal at this step is to learn a function $\mathcal{H} : \boldsymbol{A}, \boldsymbol{X} \to \boldsymbol{H} \in [0,1]^{n \times l_c}$ which maps the node attributes and graph structure to a pseudo label space represented as $\boldsymbol{H} \in [0,1]^{n \times l_c}$. For this step, we adopt the unsupervised convolution-based clustering framework of Kamhoua et al. (2022) who did an analysis on the framework of Zhang et al. (2019b). These two convolution-based clustering frameworks were empirically shown to effectively map the node attributes and graph structure to the pseudo-label space (as seen by the state-of-the-art clustering accuracy achieved by these frameworks, see more in Appendix D.2).

The frameworks of Zhang et al. (2019b) and Kamhoua et al. (2022) consisted of building the normalized symmetric graph Laplacian as $\boldsymbol{L}_{sym} = \boldsymbol{I} - \boldsymbol{D}^{\frac{-1}{2}} \boldsymbol{A} \boldsymbol{D}^{\frac{-1}{2}} = \boldsymbol{I} - \hat{\boldsymbol{A}}_{sym}$, where $\boldsymbol{I}$ is the identity matrix, $\boldsymbol{D}$ is the diagonal matrix of node degrees, and $\hat{\boldsymbol{A}}_{sym} = \boldsymbol{D}^{\frac{-1}{2}} \boldsymbol{A} \boldsymbol{D}^{\frac{-1}{2}}$ is the normalized adjacency. Using this laplacian to build a low-pass filter $\mathcal{N}(\boldsymbol{L}_{sym}) = (\boldsymbol{I} - \alpha \boldsymbol{L}_{sym})^k$ (i.e., a filter that decays with the eigenvalues of the laplacian, specifically it gets smaller for larger eigenvalues, where the rate of decay is determined by the parameters $k$ and $\alpha$). Using this filter to do a convolution on the node attributes by the left matrix multiplication $\boldsymbol{F} = \mathcal{N}(\boldsymbol{L}_{sym}) \boldsymbol{X} \in$

$\mathbb{R}^{n \times d}$. Finally using these convoluted features to minimize the clustering objective $\min_{\boldsymbol{H}} ||\boldsymbol{F}\boldsymbol{F}^T - \boldsymbol{H}\boldsymbol{H}^T||_F^2$, where $\boldsymbol{H} \in [0, 1]^{n \times l_c}$ is a pseudo label space (the cluster space in their work), and $|| \cdot ||_F$ is the Frobenius norm. They also propose a criterion for the automatic selection of $k$ and $\alpha$ (we refer the readers to their work for more details).

## 4.2 Key nodes selection as labeled nodes (KL)

The second phase in our proposed pipeline is to select a label node set $\mathcal{V}_L$ to use for sparse labels classification. Here we chose a number of nodes $l_H$ from the pseudo label class returned by the Label distribution estimation model. For every pseudo-label class $\boldsymbol{H}_{:,j}$ (where $\boldsymbol{H}_{:,j}$ is a column of $\boldsymbol{H}$), the $l_H$ nodes were chosen to be the nodes with the smallest loss used by the Label distribution estimation model (such as the loss $||\boldsymbol{F}\boldsymbol{F}^T - \boldsymbol{H}\boldsymbol{H}^T||$). As the number of nodes needed for training increased beyond the number of clusters $l_c$, the rest of the labeled nodes $l_R$ selected for training are selected at random. As such the total number of labeled nodes used for training is $|\mathcal{V}_{\mathcal{L}}| = l = l_H + l_R$.

## 4.3 Label distribution incorporation in sparse labels classification

Here we attempt to incorporate the pseudo-label distribution $\boldsymbol{H}$ in the sparse labels classification problem. To this end, we consider the sparse labels classification problem formulated via the popular label propagation (LP) (Delalleau et al., 2005). First, given extremely few labeled nodes $\mathcal{V}_L$, the label distribution $\mathcal{Y}_{\mathcal{V}_L}$ of the labeled nodes can be represented as the matrix $\boldsymbol{Y} \in [0, 1]^{n \times l}$. Each row $\boldsymbol{Y}_{i,:}$ of $\boldsymbol{Y}$ is either $\boldsymbol{e}^{(t)}$ the label indicator $[0, \dots, 0, 1, 0, \dots, 0] \in \mathbb{R}^l$ with a 1 at position $t$ if node $v_i \in \mathcal{V}_L$ and $v_i$ is labeled with its label being $t$, or $\boldsymbol{Y}_{i,:}$ is the zero vector $\vec{\boldsymbol{0}} \in \mathbb{R}^l$ if the node $v_i$ is not labeled. Second, given the graph laplacian $\boldsymbol{L}_{sym}$ and the node features $\boldsymbol{X}$, the sparse label propagation problem, consists of recovering the labels for all the unlabeled nodes by solving:

$$\min_{\boldsymbol{R}} \mu ||\boldsymbol{R} - \boldsymbol{Y}||_F^2 + \text{Tr}(\boldsymbol{R}^T \boldsymbol{L}_{sym} \boldsymbol{R}), \tag{1}$$

where the matrix $\boldsymbol{R}$ contains the recovered labels for unlabeled nodes as well as the original labels for labeled nodes, $\text{Tr}(\cdot)$ is the matrix trace operator, and $\mu$ is a user-defined hyper-parameter.

LP in equation 1 however performs poorly on SLNC, since in this setting only extremely few labels are available (see figures 1 and 4(a)). So to address SLNC with LP, we incorporate the label distribution into it via the pseudo labels $\boldsymbol{H}$. It is not straightforward to use $\boldsymbol{H}$ because in general, the order of the labels in $\boldsymbol{H}$ does not match the label order in $\boldsymbol{Y}$ i.e., the class $C_i$ in $\boldsymbol{Y}$ may be the class $C_j$ in $\boldsymbol{H}$ since $\boldsymbol{H}$ was obtained in an unsupervised manner via clustering. To solve this problem, we use the notion of neighborhood over $\boldsymbol{H}$ (one way of building such a neighborhood graph $\mathcal{G}_{\boldsymbol{H}}$ is via the outer product $\boldsymbol{A}_{\mathcal{G}_{\boldsymbol{H}}} = \boldsymbol{H}\boldsymbol{H}^T$, and setting ). We then build the normalized symmetric Laplacian $\boldsymbol{L}_{\mathcal{G}_{\boldsymbol{H}}}$ of this graph $\mathcal{G}_{\boldsymbol{H}}$ and impose that the recovered labels $\boldsymbol{R}$ should also be smooth on the graph $\mathcal{G}_{\boldsymbol{H}}$ as:

$$\min_{\boldsymbol{R}} \mu ||\boldsymbol{R} - \boldsymbol{Y}||_F^2 + \mathcal{W}(\boldsymbol{L}_{sym}) + \mathcal{K}(\boldsymbol{L}_{\mathcal{G}_{\boldsymbol{H}}}), \tag{2}$$

where the term $\mathcal{W}(\boldsymbol{L}_{sym}) = \beta_1 \text{Tr}(\boldsymbol{R}^T \boldsymbol{L}_{sym} \boldsymbol{R})$ enforces smoothness over the node neighborhood, and the term $\mathcal{K}(\boldsymbol{L}_{\mathcal{G}_{\boldsymbol{H}}}) = \beta_2 \text{Tr}(\boldsymbol{R}^T \boldsymbol{L}_{\mathcal{G}_{\boldsymbol{H}}} \boldsymbol{R})$ enforces smoothness over the pseudo label distribution.

To mitigate the effect of some possible erroneous neighborhoods in the pseudo-labels space (which will cause nodes with different true labels to have similar features), we further enforce smoothness on the real label space of the extremely few labeled nodes $\mathcal{V}_L$, by using the symmetric normalized Laplacian $\boldsymbol{L}_{\mathcal{G}_{\boldsymbol{Y}}}$ of the graph $\mathcal{G}_{\boldsymbol{Y}}$ with adjacency $\boldsymbol{A}_{\mathcal{G}_{\boldsymbol{Y}}} = \boldsymbol{Y}\boldsymbol{Y}^T$. Equation 2 will thus become:

$$\min_{\boldsymbol{R}} \mu ||\boldsymbol{R} - \boldsymbol{Y}||_F^2 + \mathcal{W}(\boldsymbol{L}_{sym}) + \mathcal{K}(\boldsymbol{L}_{\mathcal{G}_{\boldsymbol{H}}}) + \mathcal{P}(\boldsymbol{L}_{\mathcal{G}_{\boldsymbol{Y}}}), \tag{3}$$

where $\mathcal{P}(m\boldsymbol{L}_{\mathcal{G}_{\boldsymbol{Y}}}) = \beta_3 \text{Tr}(\boldsymbol{R}^T \boldsymbol{L}_{\mathcal{G}_{\boldsymbol{Y}}} \boldsymbol{R})$. One can verify that the solution to equation 3 is:

$$\boldsymbol{R} = \mu(\mu\boldsymbol{I} + \boldsymbol{L}_A)^{-1}\boldsymbol{Y}, \tag{4}$$



Figure 2: Illustration of the optimization strategy on Cora (see Section 4.4). The first plot shows the adjacency $\boldsymbol{A}_{\mathcal{G}_H}$ of the graph $\mathcal{G}_H$ without optimization. The second plot is the normalized adjacency $\hat{\boldsymbol{A}}_{sym}$ of graph $\mathcal{G}_A$. The last plot is the optimized adjacency $\boldsymbol{A}_{\mathcal{G}_H}$. The yellow points represent edges, showing that the adjacencies $\hat{\boldsymbol{A}}_{sym}$ and the optimized $\boldsymbol{A}_{\mathcal{G}_H}$ are both sparse.

where $\boldsymbol{L}_A = \beta_1 \boldsymbol{L}_{sym} + \beta_2 \boldsymbol{L}_{\mathcal{G}_H} + \beta_3 \boldsymbol{L}_{\mathcal{G}_Y}$ is a weighted Laplacian (see Appendix B.1). Finally, to simplify our framework we set $\beta_1 = \beta_2 = \beta_3 = \frac{1}{3}$, so that $\boldsymbol{L}_A = \frac{1}{3}(\boldsymbol{L}_{sym} + \boldsymbol{L}_{\mathcal{G}_H} + \boldsymbol{L}_{\mathcal{G}_Y})$ becomes simply the average of laplacians.

## 4.4 OPTIMIZING LABEL DISTRIBUTION INCORPORATION (KG)

The solution 4 can be approximated via the matrix multiplication:

$$\boldsymbol{R}^{t+1} = (1 - \alpha)\boldsymbol{Y} + \alpha \hat{\boldsymbol{A}}_A \boldsymbol{Y}^t, \tag{5}$$

where $\hat{\boldsymbol{A}}_A = \boldsymbol{I} - \boldsymbol{L}_A = \frac{1}{3}(\hat{\boldsymbol{A}}_{sym} + \hat{\boldsymbol{A}}_{\mathcal{G}_H} + \hat{\boldsymbol{A}}_{\mathcal{G}_Y})$ is the average of the normalized adjacencies of the graphs, and $\alpha = \frac{\mu}{1+\mu}$.

In general, the normalized adjacency $\hat{\boldsymbol{A}}_{sym}$ is sparse as shown in figure 2. This is due to the fact that the nodes in real-world attributed graphs $\mathcal{G}_A$ are usually loosely connected and only very popular nodes may have a higher connectivity and thus a higher node degree. Moreover, the normalized adjacency $\hat{\boldsymbol{A}}_{\mathcal{G}_Y}$ will also be sparse since only very few nodes are used as labeled nodes for training.

However, building the cluster graph adjacency as $\boldsymbol{A}_{\mathcal{G}_H} = \boldsymbol{H}\boldsymbol{H}^T$ will result in a very dense graph as shown in the first plot in figure 2. This will cause the solution 5 to be slow as the matrices to be multiplied will be dense. Moreover, doing so may not well capture errors in clustering, and may also not account for links between clusters. To solve these problems, we build the adjacency $\boldsymbol{A}_{\mathcal{G}_H}$ by building a KNN Graph from the Singular Value Decomposition (SVD) of $\boldsymbol{F}$ (See Appendices C and C.1 for more details). We set a high number of neighbors for the KNN graph to capture the relationship between different clusters as well as within the same cluster (we use 60 in our experiments, see Section D.1 for sensitivity studies).

## 4.5 GENERALIZING THE PROPOSED FRAMEWORK

Fu et al. (2020) showed that the graph convolution $\mathcal{N}(\boldsymbol{L}_{sym})\boldsymbol{X}$ is an approximation to the solution of the sparse label propagation problem 1 over node attributes rather than labels (commonly known as feature denoising, or graph signal denoising). Following his proof, one can equally show that the graph convolution $\mathcal{N}(\boldsymbol{L}_A)\boldsymbol{X}$ using the average laplacian $\boldsymbol{L}_A$ instead of the graph laplacian $\boldsymbol{L}_{sym}$ is an approximation to the solution to the feature denoising problem over the graph structure, pseudo label, and key labels spaces. As such, we generalize solutions to the sparse labels classification problem to GNN models such as SGC (Wu et al., 2019) and any other GNN framework, by using $\boldsymbol{A}_A$ as the normalized graph adjacency. In this work, we will use SGC which aims to minimize $\mathcal{Z}(\mathcal{F}_\theta(\mathcal{N}(\boldsymbol{L}_A)\boldsymbol{X}), \mathcal{Y}_{\mathcal{V}_l})$, where $\mathcal{Z}(\cdot)$ is any suitable classification loss.

Please find our algorithms and time complexity analysis for (a) ELI 1, and (b) for Label distribution estimation and Key labeled node selection 2 in Appendix C.

## 5 EXPERIMENTS

In this section, we report and validate the effectiveness of our proposed framework (ELI) and its generalization for SLNC. The code is attached and will be released later on Git Hub.

## 5.1 DATASETS

We used 7 benchmark attributed graphs to validate the SLNC performance of ELI and its generalization. We used: (1) the citation networks Cora (Yang et al., 2016), Citeseer (Giles et al., 1998), and Pubmed (Gasteiger et al., 2019), (2) the web page graph Wiki (Zhang et al., 2019b), (3) the co-purchase networks Computers (McAuley et al., 2015), and Photos (Ma et al., 2021), as well as (4) the coauthor graph Cs (Fey & Lenssen, 2019). Statistics of the datasets can be found in Table 2.

Table 2: Statistics of the datasets.

| Dataset | #Node | #Edge | #Attribute | #Class |
|---|---|---|---|---|
| Cora | 2708 | 5429 | 1433 | 7 |
| Citeseer | 3327 | 4732 | 3703 | 6 |
| Pubmed | 19717 | 44338 | 500 | 3 |
| Wiki | 2405 | 17981 | 4973 | 17 |
| Computers | 13752 | 491722 | 765 | 10 |
| Photo | 7650 | 238162 | 745 | 8 |
| Cs | 18333 | 163788 | 6805 | 15 |

## 5.2 EVALUATION METRICS

For comparison and evaluation, we visualize the normalized classification accuracy (normalized to be in [0,1]) for all semi-supervised models against the total number of labels used for training. Each experiment is run 10 times and the mean and standard deviations of the accuracy are plotted.

For the labels used, we randomly selected #num $\times c$ training nodes over the entire set of nodes $\mathcal{V}$, where $c$ is the number of unique classes and #num was varied from $1 - 4$. More experiments can be found in Appendix D, where we (a) use #num from $1 - 20$, and (b) also use the setting where training nodes are selected on a per-class basis rather than over the entire node set.

## 5.3 BASELINES

For comparison, we will use (a) our proposed framework (ELI) as in Algorithm 1 both with the Label Propagation approach by setting $w = 0$ which we term LP-ELI and our generalization using SGC by setting $w! = 0$ which we term SGC-ELI. (b) The semi-supervised models: Label Propagation (LP) (Delalleau et al., 2005), and SGC (Wu et al., 2019) without ELI. (c) The contrastive based network CGPN Wan et al. (2021) proposed for the sparse label setting. (d) Last but not least, we use two pre-training bases lines DGI (Veličković et al., 2019), and GMI (Peng et al., 2020). where we also do not pair them with ELI.

## 5.4 SET-UP

DGI and GMI were pre-trained on a Linux server with Python(Van Rossum & Drake Jr, 1995) version 3.9.15, PyTorch(Paszke et al., 2019) version 1.13.1, and TensorFlow (Abadi et al., 2015) version 2.11. The server was equipped with an RTX 3090 and Intel(R) Xeon(R) Silver 4114 CPU @ 2.20GHz, and 252G of RAM. While all other downstream tasks were done on a Windows 11 machine with an Intel(R) Core(TM) i5-13500 @ 2.50GHz, a 2080Ti, and 32GB of RAM.

| Model | Cora | | Citeseer | | Wiki | | Pubmed | |
|---|---|---|---|---|---|---|---|---|
| | #1 | #2 | #1 | #2 | #1 | #2 | #1 | #2 |
| SGC | $38.58 \pm 6.53$ | $49.70 \pm 10.16$ | $30.97 \pm 6.91$ | $34.55 \pm 7.47$ | $40.97 \pm 6.09$ | $52.37 \pm 4.91$ | $40.37 \pm 7.78$ | $44.54 \pm 10.52$ |
| LP | $41.47 \pm 6.79$ | $49.45 \pm 7.11$ | $30.45 \pm 6.64$ | $31.29 \pm 5.92$ | $31.18 \pm 4.85$ | $36.74 \pm 5.02$ | $43.24 \pm 9.72$ | $52.73 \pm 7.28$ |
| CGPN | $17.95 \pm 6.12$ | $21.84 \pm 6.81$ | $16.85 \pm 4.83$ | $21.85 \pm 3.52$ | $09.15 \pm 4.23$ | $11.88 \pm 7.12$ | - | - |
| DGI | $48.08 \pm 8.98$ | $59.67 \pm 7.51$ | $40.35 \pm 9.48$ | $42.39 \pm 6.35$ | $33.19 \pm 3.61$ | $41.37 \pm 4.78$ | $44.84 \pm 10.09$ | $51.36 \pm 10.17$ |
| GMI | $50.87 \pm 4.76$ | $58.01 \pm 6.50$ | $45.31 \pm 8.74$ | $44.61 \pm 6.38$ | $30.78 \pm 3.61$ | $38.66 \pm 4.58$ | $45.15 \pm 8.86$ | $51.72 \pm 8.73$ |
| LP-ELI | $69.72 \pm 0.00$ | $69.14 \pm 1.93$ | $66.21 \pm 0.00$ | $64.99 \pm 1.94$ | $51.26 \pm 0.00$ | $54.35 \pm 1.78$ | $68.08 \pm 0.00$ | $65.59 \pm 2.96$ |
| SCG-ELI | $67.16 \pm 0.00$ | $68.90 \pm 2.76$ | $64.77 \pm 0.00$ | $61.02 \pm 3.10$ | $56.12 \pm 0.00$ | $59.95 \pm 2.39$ | $66.24 \pm 0.00$ | $62.07 \pm 7.53$ |

Table 3: Tabular results of $\%$ classification accuracy on Cora, Citeseer, Wiki, and Pubmed for different models using 1 label (#1) and 2 labels (#2).

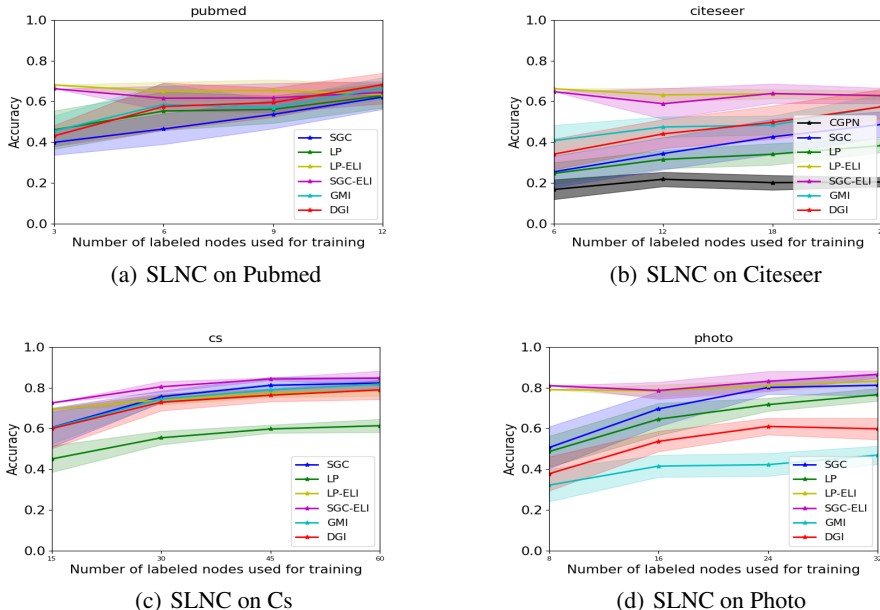

Figure 3: Comparison between ELI-enhanced models and other baselines for SLNC on Pubmed, Citeseer, and Cs. See parameter settings in Section 5.5.

| Model | Cs | | Photo | | Computers | |
|---|---|---|---|---|---|---|
| | #1 | #2 | #1 | #2 | #1 | #2 |
| SGC | $62.85 \pm 7.79$ | $70.88 \pm 5.03$ | $56.06 \pm 7.91$ | $71.28 \pm 7.28$ | $53.50 \pm 9.81$ | $63.94 \pm 3.38$ |
| LP | $46.80 \pm 6.58$ | $53.93 \pm 3.22$ | $53.03 \pm 9.97$ | $63.54 \pm 7.61$ | $50.18 \pm 6.79$ | $59.96 \pm 3.37$ |
| CGPN | - | - | - | - | - | - |
| DGI | $63.36 \pm 6.74$ | $69.94 \pm 5.14$ | $40.54 \pm 10.12$ | $51.31 \pm 6.98$ | $41.63 \pm 9.09$ | $49.66 \pm 2.66$ |
| GMI | $62.74 \pm 6.03$ | $71.19 \pm 5.47$ | $32.41 \pm 8.21$ | $38.65 \pm 6.52$ | $40.03 \pm 10.06$ | $47.44 \pm 3.03$ |
| LP-ELI | $69.40 \pm 0.00$ | $75.43 \pm 2.60$ | $79.00 \pm 0.00$ | $77.69 \pm 3.48$ | $71.17 \pm 0.00$ | $70.40 \pm 1.45$ |
| SCG-ELI | $72.50 \pm 0.00$ | $82.62 \pm 3.19$ | $80.97 \pm 0.00$ | $82.03 \pm 2.82$ | $70.78 \pm 0.00$ | $71.96 \pm 1.71$ |

Table 4: Tabular results of % classification accuracy on Cs, Photo, and Computers for different models using 1 label (#1) and 2 labels (#2).

## 5.5 PARAMETER SETTINGS

### 5.5.1 SEMI-SUPERVISED MODELS

For the LP(Delalleau et al., 2005) model, we followed the implementation of Huang et al. (2021) and set the maximum propagation power to 60, the p parameter to 0.6, and the $\alpha$ parameter to 0.5. For SGC Wu et al. (2019), we set the convolution power $k$ to 5 and use a multi-layer perceptron (MLP) from scikit-learnPedregosa et al. (2011) with the solver set as adam, alpha as 1e-3, hidden_layer_sizes as (16,), activation as logistic, random_state as 0, and learning_rate_init=0.01. For CGPN, we followed the original paper settings.

### 5.5.2 PRE-TRAINING MODELS

On DGI (Veličković et al., 2019) we follow the original paper settings, where we set the number of epochs to 10000, the patience to 20, the dropout probability to 0, the learning rate to 0.001, the l2 regularization to 0, and we set the output feature dimensions to 512 for all datasets except for the larger datasets (i.e., Pubmed, Computers, Photo, Cs) for which we use 256 instead. We equally follow the original paper settings for GMI (Peng et al., 2020), where we set the number of epochs to 550, the patience to 20, the dropout probability to 0, the learning rate to 0.001, the l2 regularization to 0, the number of negative samples to 5, the alpha parameter for the loss to 0.8, the beta parameter for the loss to 1.0, the gamma parameter for the loss to 1.0, and we set the output feature dimensions to 256 for all datasets. All pre-training was done on the RTX 3090. For the downstream classification tasks, we

used these pre-trained features in a multi-layer perceptron model (MLP) from scikit-learn(Pedregosa et al., 2011) with the same parameter settings as the one used for the semi-supervised SGC model discussed in Section 5.5.1. We used this MLP instead of the simpler Logistic Regression as proposed in the original papers because the MLP performed much better.

### 5.5.3 ELI

For Label Distribution Estimation in steps 2-4 of Algorithm 1, we use the variant of attributed graph clustering via adaptive graph convolution (AGC) from Kamhoua et al. (2022). For all datasets, we set the tolerance for their stopping criteria to 0.001, and the maximum number of iterations $k$ to 60. For all datasets, no normalization is used for the features (due to space, see clustering results in Appendix C). We then incorporate this into the LP or SGC models (following Section 4) depending on $w$ in Algorithm 1, we name these SGC-ELI and LP-ELI. The SGC-ELI and LP-ELI models here use the same $k$, $p$, and power parameters used for the SGC and LP in Section 5.5.1.

### 5.6 COMPARISON ANALYSIS

Here, we compare and contrast the SLNC performance of baselines enhanced by ELI (i.e., SGC-ELI and LP-ELI) against the performance of baselines without ELI (i.e., SGC, LP, DGI, and GMI).

First, it can be observed from figures 1 and 3 and tables 3 and 4 that GNNs enhanced by ELI (i.e., SGC-ELI and LP-ELI) outperformed the non-enhanced baselines by 10-20% when the total number of labels used for training was extremely small. However, as the total number of labels kept growing, this margin converged and narrowed down (see more extensive plots in Appendix D). Moreover, it can also be observed that SGC-ELI often outperformed LP-ELI, which can be explained by the fact that the function $\mathcal{F}_\Theta(\boldsymbol{A}, \boldsymbol{X})$ learned by SGC-ELI via the graph convolution and the neural network helps SGC-ELI not only to account for graph structure and labels but also the node features.

On the other hand, there seems to be a trade of blow between, the pre-trained baselines (DGI, and GMI) and the non-pre-trained baselines (SGC, and LP). This is as expected as the pre-trained baselines just like the non-pre-trained baselines will both learn only structural and/or node information while not explicitly learning about the label distribution of the graph, and as such in the presence of extremely few randomly selected labels for training, both -pre-trained and non-pre-trained baselines will not have enough guidance to learn a good $\mathcal{F}_\Theta(\boldsymbol{A}, \boldsymbol{X})$.

Lastly, it can also be observed that though CGPN was proposed for a setting with extremely few labels, it heavily relied on what nodes were selected per-class, and as such it struggles in the SLNC setting where nodes may be randomly selected not on a per-class basis. Furthermore, this model took a very long time to run on our downstream intel 5 13500 machine equipped with the RTX 2080Ti. For example, on Citeseer it took more than 48 seconds per run per number of labels used compared to 0.27 seconds for LP-ELI or 2.18 seconds for SGC-ELI. On larger datasets like Pubmed and Photo, it was still running even after 45 mins (so we stopped it and did not use it on the larger Pubmed, Cs, Computers and Photo). For ablation and sensitivity analysis, we refer the reader to Appendix D.1.

## 6 CONCLUSION AND LIMITATIONS

In this work, we presented a real-world scenario classification task in which we have only very few randomly chosen node labels named Sparse Labels Node Classification (SLNC). We then presented a framework that performs about 10-20% better than traditional baselines for SLNC. Finally, we conducted ablation and sensitivity studies on the proposed framework. A key limitation of our work is that, though we tried to simulate a real-world scenario where we randomly select only a few labels (not on a per-class basis), we still required the total number of possible unique classes to be known in advance which is generally not the case in real life. There are a few possible future directions, which include but are not limited to; exploring the ELI framework or variants of it for other domains (such as images, videos, shapes, and others), overcoming the main limitation of this work by generalizing ELI to scenarios where the total number of possible unique classes are not known in advance, further generalizing and optimizing ELI by varying the model used for the label distribution estimation (Section 4.1), or even the way the label it is incorporated in SLNC (Section 4.3). We believe this work is beneficial as it can help the industry to alleviate huge costs and time used in labeling.

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

# Sparse Labels Node Classification: Unsupervised Learning for Mentoring Supervised Learning in Sparse Label Settings.
# Appendices:

## A MORE ON LIMITATIONS OF THE PROPOSED FRAMEWORK

Although a key limitation of our work is that we require prior knowledge of the number of all unique classes in the graph (see Section 6), we want to note here that our framework can also be used in cases where this is not known. We simply guess this number of possible unique classes and then use this guess to generate the pseudo-labels in the first stage of our framework (see step 2 in Algorithm 1) and then use these pseudo-labels in the next stages of our framework Algorithm 1. Though we did not experiment this setting, it is a good possible future extension and research direction of the current work.

A second limitation of this current work is that due to time constraints, we did not use more varied GNN models with our proposed generalization of the framework (see Section 4.5). This will indeed be very interesting to see in possible future extensions as well as to see how general is the generalization (i.e., how well it works with more varied models).

A last possible limitation we perceive is the fact that we did not explore other tasks besides node classification in the proposed sparse labels classification setting. It will indeed be a good future direction to explore other tasks in this setting as well as see how similar frameworks or improved frameworks can be implemented for those tasks since in real life the sparse label setting is ubiquitous.

## B MORE ON THE SPARSE LABELS CLASSIFICATION PROBLEM

### B.1 SOLUTION TO THE PROPOSED FRAMEWORK

In this section, we derrive the solution to the proposed framework. Recall that the proposed framework is of the form:

$$\min_{\boldsymbol{R}} \mu||\boldsymbol{R} - \boldsymbol{Y}||_F^2 + \mathcal{W}(\boldsymbol{L}_{sym}) + \mathcal{K}(\boldsymbol{L}_{\mathcal{G}_{\boldsymbol{H}}}) + \mathcal{P}(\boldsymbol{L}_{\mathcal{G}_{\boldsymbol{Y}}}), \tag{6}$$

setting $\mathcal{W}(\boldsymbol{L}_{sym}) + \mathcal{K}(\boldsymbol{L}_{\mathcal{G}_{\boldsymbol{H}}}) + \mathcal{P}(\boldsymbol{L}_{\mathcal{G}_{\boldsymbol{Y}}}) = \mathcal{Q}(\boldsymbol{L}_A)$, we can rewrite 6 as

$$\min_{\boldsymbol{R}} ||\boldsymbol{R} - \boldsymbol{Y}||_F^2 + \mathcal{Q}(\boldsymbol{L}_A), \tag{7}$$

where also, $\mathcal{Q}(\boldsymbol{L}_A) = \mathbf{Tr}(\beta_1 \boldsymbol{R}^T \boldsymbol{L}_{sym} \boldsymbol{R} + \beta_2 \boldsymbol{R}^T \boldsymbol{L}_{\mathcal{G}_{\boldsymbol{H}}} \boldsymbol{R} + \beta_3 \boldsymbol{R}^T \boldsymbol{L}_{\mathcal{G}_{\boldsymbol{Y}}} \boldsymbol{R})$ Taking, simplifying, and setting the derivative of equation 7 w.r.t $\mathbf{R}$ to 0 we have:

$$2\mu(\boldsymbol{R} - \boldsymbol{Y}) + 2(\boldsymbol{R}\boldsymbol{L}_A) = 2(\mu\boldsymbol{I} + \boldsymbol{L}_A)\boldsymbol{R} - \mu\boldsymbol{Y} = 0, \tag{8}$$

where $\boldsymbol{L}_A = \beta_1 \boldsymbol{L}_{sym} + \beta_2 \boldsymbol{L}_{\mathcal{G}_{\boldsymbol{H}}} + \beta_3 \boldsymbol{L}_{\mathcal{G}_{\boldsymbol{Y}}}$ is a weighted Laplacian. Thus we have the solution being:

$$\boldsymbol{R} = \mu(\mu\boldsymbol{I} + \boldsymbol{L}_A)^{-1}\boldsymbol{Y}, \tag{9}$$

### B.2 APPROXIMATING THE SOLUTION AND GENERALIZING

Given that the matrix inversion in equation 9 is very expensive. As such, it is generally approximated. One can rewrite where $\boldsymbol{L}_A = \boldsymbol{I} - \hat{\boldsymbol{A}}_A$ where $\hat{\boldsymbol{A}}_A = \frac{1}{3}(\hat{\boldsymbol{A}}_{sym} + \hat{\boldsymbol{A}}_{\mathcal{G}_{\boldsymbol{H}}} + \hat{\boldsymbol{A}}_{\mathcal{G}_{\boldsymbol{Y}}})$, given that we set

$\beta_1 = \beta_2 = \beta_3 = \frac{1}{3}$. $\hat{A}_A$ can be seen as an average symmetrically normalized adjacency (note that given an adjacency matrix $A$ and the diagonal matrix $D$ of node degrees, one can obtain the average symmetrically normalized adjacency as $\hat{A} = D^{\frac{-1}{2}} A D^{\frac{-1}{2}}$. Using the symmetrically normalized adjacency, one can rewrite the solution is equation 9 as:

$$\boldsymbol{R} = \mu(\mu\boldsymbol{I} + (\boldsymbol{I} - \hat{\boldsymbol{A}}_A))^{-1}\boldsymbol{Y} = \frac{\mu}{1+\mu}(\boldsymbol{I} - \frac{1}{1+\mu}\hat{\boldsymbol{A}}_A)^{-1}\boldsymbol{Y},$$

setting $\alpha = \frac{1}{1+\mu}$, we have that:

$$\boldsymbol{R} = (1-\alpha)(\boldsymbol{I} - \alpha\hat{\boldsymbol{A}}_A)^{-1}\boldsymbol{Y} = (1-\alpha)\sum_{k=0}^{\infty}(\alpha\hat{\boldsymbol{A}}_A)^k\boldsymbol{Y}.$$

As such we can approximate the solution in equation 9 by performing until convergence the iterative update:

$$\mathbf{R}^{t+1} = (1-\alpha)\mathbf{Y} + \alpha\hat{\mathbf{A}}_A\mathbf{Y}^t. \tag{10}$$

To generalize this framework to other GNNs, we recall that the graph signal denoising problem is of the form:

$$\min_{\boldsymbol{P}} ||\boldsymbol{P} - \boldsymbol{X}||_F^2 + \alpha_1\mathcal{W}(\boldsymbol{L}_{sym}), \tag{11}$$

where the term $||\boldsymbol{P} - \alpha\boldsymbol{X}||_F^2$ tries to make the learned denoised features $\boldsymbol{P}$ to be as close as possible to the noisy ones features $\boldsymbol{X}$, and the term $\mathcal{W}(\boldsymbol{L}_{sym}) = \mathbf{Tr}(\boldsymbol{X}^T\boldsymbol{L}_{sym}\boldsymbol{X})$ tries to make neighbors on the graph to have similar denoised features. For SLNC, one will need to incorporate some degree on information about the label distribution into equation 11. Doing so will imply trying to enforce that the learned denoised featured of nodes sharing similar pseudo-labels (as in Sections 4.1) are equally similar. This can be enforced by adding the two terms $\mathcal{K}(\boldsymbol{L}_{\mathcal{G}_H}) = \mathbf{Tr}(\alpha_2\boldsymbol{P}^T\boldsymbol{L}_{\mathcal{G}_H}\boldsymbol{P})$, and $\mathcal{P}(\boldsymbol{L}_{\mathcal{G}_Y}) = \mathbf{Tr}(\alpha_3\boldsymbol{P}^T\boldsymbol{L}_{\mathcal{G}_Y}\boldsymbol{P}))$ into equation 11, thus yielding:

$$\min_{\boldsymbol{P}} \mu||\boldsymbol{P} - \boldsymbol{X}||_F^2 + \mu\mathcal{Q}(\boldsymbol{L}_A), \tag{12}$$

where $\mathcal{Q}(\boldsymbol{L}_A) = \mathbf{Tr}(\alpha_1\boldsymbol{P}^T\boldsymbol{L}_{sym}\boldsymbol{P} + \alpha_2\boldsymbol{P}^T\boldsymbol{L}_{\mathcal{G}_H}\boldsymbol{P} + \alpha_3\boldsymbol{P}^T\boldsymbol{L}_{\mathcal{G}_Y}\boldsymbol{P})$ and the terms.

Setting $\alpha_1 = \alpha_2 = \alpha_3 = \frac{1}{3}$ and following similar steps in equations 8 and 9, we have that the solution to equation 12 is :

$$\boldsymbol{P} = (\boldsymbol{I} + \mu\boldsymbol{L}_A)^{-1}\boldsymbol{X}, \tag{13}$$

where $\boldsymbol{L}_A = \boldsymbol{I} - \hat{\boldsymbol{A}}_A$, and $\hat{\boldsymbol{A}}_A = \frac{1}{3}(\hat{\boldsymbol{A}}_{sym} + \hat{\boldsymbol{A}}_{\mathcal{G}_H} + \hat{\boldsymbol{A}}_{\mathcal{G}_Y})$. One can observe that $(\boldsymbol{I} + \mu\boldsymbol{L}_A)^{-1}\boldsymbol{X}$ in equation 13 is a graph convolution with the grap convolution filter being the lowpass filter $(\boldsymbol{I} + \mu\boldsymbol{L}_A)^{-1}$. As such, one can use different convolution functions with ELI as well by just using the average laplacian $\boldsymbol{L}_A$.

## C   ALGORITHM FOR ELI AND THE CHOSEN FRAMEWORK FOR PSEUDO-LABELS APPROXIMATION

In this section, we show the adaptation of the clustering algorithm of Kamhoua et al. (2022) that we used to generate the KNN graph (see Sections 4.1 and 4.3), as well as to select key labels (see Section 4.2)

---

**Algorithm 1** : ELI (A Proposed Framework) for SLNC

---

1. **input** Graph adjacency $\boldsymbol{A}$, node attributes $\boldsymbol{X}$, maximum iteration for pseudo-label estimation $k$, a flag $w$ for which learning method to choose to solve the sparse labels classification problem, and parameters of the learning method.

2. Obtain both the label distribution $\boldsymbol{H}$ and the $l_H$ key labeled nodes from the pseudo label space (as in Sections 4.3, 4.4 and 4.2) using Algorithm 2

4. Build the real label graph adjacency $\boldsymbol{A}_{\mathcal{G}_Y} = \boldsymbol{Y}\boldsymbol{Y}^T$ (as in Section 4.3)

**if** $w = 0$ **then**

    5a. Solve the sparse labels classification problem using the Label propagation framework as in equation 5

**else**

    5b. Solve the sparse labels classification problem using the generalization of the framework as in Section 4.5 (in this work we use SGC here).

**end if**

6. **return** the labels $\mathbf{R}$, and use the set of labels corresponding to those that were unlabeled at training for evaluation.

---

**Algorithm 2** AGC: Adaptive Graph Convolution based Clustering algorithm

---

1: **Input:** Adjacency matrix $\boldsymbol{A}$ and attribute matrix $\boldsymbol{X}$ of graph $\mathcal{G}$, number of clusters $K$, graph filtering rate $\alpha$, maximum iteration number $L_{\max}$, tolerance value $tol$, number of nearest neighbors $nn$

2: **Output:** A set of clusters $C$, the KNN-graph adjacency $\boldsymbol{A}_{\mathcal{G}_H}$, list of nodes $l_H$ with min clustering loss per cluster

3: Set $\boldsymbol{A}_{ii} = 1$ for all nodes $v_i \in \mathcal{V}$ of graph $\mathcal{G}$

4: Compute symmetric graph Laplacian $\boldsymbol{L}_{sym} = \boldsymbol{I} - \boldsymbol{D}^{-1/2}\boldsymbol{A}\boldsymbol{D}^{-1/2}$

5: Initialize $\boldsymbol{Y}^{(0)} = \boldsymbol{X}$

6: **for** $l = 1$ to $L_{\max}$ **do**

7:     Perform $\boldsymbol{Y}^{(k)} = (\boldsymbol{I} - \alpha\boldsymbol{L}_{sym})\boldsymbol{Y}^{(l-1)}$

8:     obtain singular vectors $\boldsymbol{U}$ by solving $[\boldsymbol{U}, \boldsymbol{S}, \boldsymbol{V}] = SVD(\boldsymbol{Y}^{(l)})$

9:     store these singular values in a temporary $\boldsymbol{U}_{temp}$

10:     Obtain $K$ clusters $C^{(l)}$ by applying K-means on $\boldsymbol{U}$

11:     **if** $Comp(C^{(l-1)}) - Comp(C^{(l)}) \leq tol$ **then**

12:         Obtain KNN-graph adjacency $\boldsymbol{A}_{\mathcal{G}_H}$ from $\boldsymbol{U}_{temp}$ from iteration, using $nn$ nearest neighbors

13:         get $l_H$ from $C^{(l)}$

14:         **break**

15:     **end if**

16: **end for**

17: Set $C = C^{(k)}$

---

Algorithm 2 shows our modification of the framework of Kamhoua et al. (2022). The stopping metric that measures the compactness of clusters is given as follows:

$$Comp(C) = \frac{1}{|C|}\sum_{k=1}^{K}\frac{1}{|C_c|(|C_k|-1)}\sum_{v_i,v_j\in C_k,v_i\neq v_j}||\boldsymbol{y}_i - \boldsymbol{y}_j||^2, \qquad (14)$$

where $C = \{C_1, \cdots, C_k\}$ is a set of clusters. Each node $v_i$ is associated with an attribute vector $\mathbf{y}_i$. For our adaptation, we set $tol$ to 0.001, $L_{max}$ to 60, and $\alpha$ to 0.5 for all datasets. Zhang et al. (2019b) and Kamhoua et al. (2021) observed that the convolution operation is beneficial for clustering as it helps to group nodes with higher connectivity while being harmful as it will cause nodes with similar features to be embedded away from each other. They further proposed $Comp(C)$ as a way of finding the best trade-off between the benefit and the harm of the convolution operation for clustering. We refer the reader to Zhang et al. (2019b) and Kamhoua et al. (2021) for an extensive study on the reason for the efficiency of the convolution operation for clustering.

**The total time complexity of the models enhanced with ELI for our experiments is as follows:**

- for LP-ELI where $w = 0$ in Algorithm 1 is $O(cN + cnd + knd + dn^2 + tcnd)$, where $cN$ is for equation (5), $O(nd + kn)$ is for the KNN graph, $O(dn^2)$ is for the SVD, and $O(tcnd) = O(nd)$ is for kmeans MacQueen et al. (1967) since t and c are extremely small.

- the total time complexity for SGC-ELI is $O(Nd + ndc + cnd + knd + dn^2 + tcnd)$, where $Nd$ is for the filtering operation $\hat{\mathbf{X}} = \mathcal{N}(\mathbf{L}_A)\mathbf{X}$ defined in section 4.5, $ndc$ for multiplying the filtered features and the neurons of size $d$ by $c$ i.e., $\hat{\mathbf{X}}\mathbf{W}$, and the rest being similar to those defined for the LP-ELI model. Giving a simplified complexity of $O(dn^2 + ndc + knd)$, with $c << d << N << n^2$ and $t$ being a constant.

### C.1 More on the KNN graph regularizer

As mentioned in Section 4.4, building the adjacency $A_{\mathcal{G}_H}$ as $A_{\mathcal{G}_H} = \boldsymbol{H}\boldsymbol{H}^T$, will result in connecting only nodes belonging to the same clusters (since $\boldsymbol{H}$ obtained from clustering are hard cluster assignments), as such will ignore possible links between clusters. Moreover, such a strategy will result in a very dense adjacency, not suitable for the convolution operation in most GNNs.

To address this, we build a KNN graph on soft cluster assignment, which will (a) not result in isolated components, and (b) address connections between clusters which will be more robust in case some nodes end up grouped in the wrong clusters, and (c) result in a sparse adjacency suitable for the convolution operation in GNNs. One can see that $\boldsymbol{Y}^{(k)} = (\boldsymbol{I} - \alpha\boldsymbol{L}_{sym})\boldsymbol{Y}^{(l-1)}$ in algorithm 1 is the same as $\boldsymbol{F}$ in Section 4.1. As such, building the KNN graph from the SVD of $\boldsymbol{F}$ in in Section 4.1 is the same as building the KNN graph on $\boldsymbol{Y}$ in algorithm 1, which means building the KNN graph on $\boldsymbol{U}$. One can notice that this is equivalent to building the KNN graph on the soft cluster assignments, where the assignments are given by $\boldsymbol{U}$.

## D More Experiments

Table 5: Ablation naming scheme

| Name | Description | Name | Description |
|---|---|---|---|
| no KG {name}-ELI | model named {name} without the KNN Graph in section 4.4 | full {name}-ELI | model with all ELI steps |
| no KL {name}-ELI | model named {name} without the Key Labels in section 4.2 | no KL no KG {name}-ELI | model no KNN graph and no Key labeled nodes |

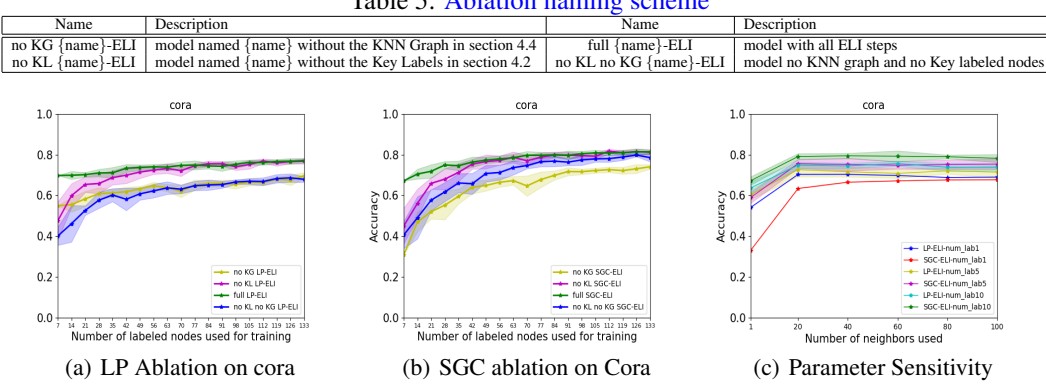

(a) LP Ablation on cora     (b) SGC ablation on Cora     (c) Parameter Sensitivity

Figure 4: Ablation studies and Parameter Sensitivity on Cora.

### D.1 Ablation and sensitivity analysis

We conducted ablation studies by comparing the performance of the ELI-augmented LP and SGC models (full SGC-ELI, full LP-ELI) against; (1) those without the key node as labeled node selection (no KL SGC-ELI, and no KL LP-ELI), but with label distribution incorporation via the KNN graph, (2) those without label distribution incorporation via the KNN graph (no KG SGC-ELI, and no KG LP-ELI), but with the key node as labeled node selection, and (3) those with no ELI (no KL no KG SGC-ELI and no KL no KG LP-ELI)

It can be observed from Figure 4 that the models augmented with the full ELI framework (full SGC-ELI, and full LP-ELI) outperform both the models with the partial ELI (no KG SGC-ELI, no KG LP-ELI, no KL SGC-ELI, and no KL LP-ELI) and the models with no ELI (no KL no KG SGC-ELI, and no KL no KG LP-ELI). Moreover, it can equally be seen that the models with partial ELI also outperform the models with no ELI. Finally, it can be seen that the models with only the label distribution incorporation step of ELI (no KL SGC-ELI, and no KL LP-ELI) outperform the ones with only the key nodes as label nodes selection (no KG SGC-ELI, no KG LP-ELI), which validates our intuition that for good performance in the presence of very sparse labels, some amount of information about the label distribution is important.

We equally conduct sensitivity analysis on the effects of the number of neighbors used for the KNN graph for the label distribution incorporation step of ELI in Sections 4.3 and 4.4, as well as varying the number of labels $l$ (num_lab in the plot) used for training. It can be seen from figure 4(c) that the SLNC performance of the models increases as the number of neighbors used increases up to a threshold beyond which further increase in the number of neighbors does not bring more increase in the SLNC performance. We thus recommend users use a high number of neighbors while trying to find a trade-off with the run time. We recommend around 60 neighbors as used for all our experiments on all datasets. Equally, increasing num_lab led to a model performance increase as expected.

### D.2  EXPERIMENTS ON THE CHOSEN FRAMEWORK FOR PSEUDO-LABELS APPROXIMATION

Figure 5 shows the clustering performance of Algorithm, 2. It can be observed that in general, this

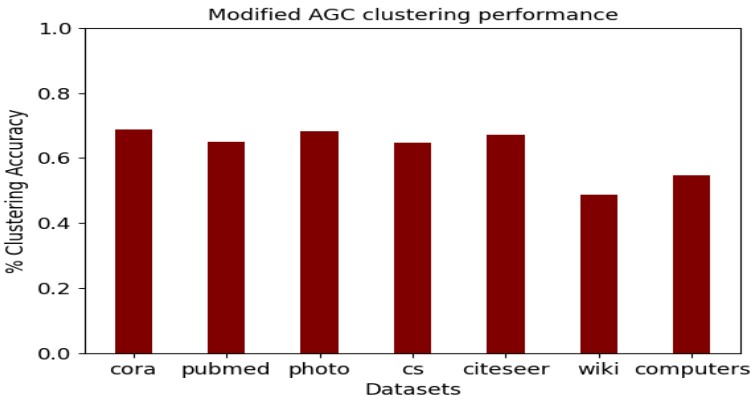

Figure 5: Clustering performance of Algorithm 2 using parameters in Appendix C.

simple framework achieves a reasonable clustering accuracy across all datasets, and as such is a good framework for estimating the label distribution in a graph since higher clustering accuracy indicates that nodes with similar classes are grouped together in a pseudo-class (cluster).

In this section, we equally investigate the patterns and correlation between the performance of ELI with the clustering performance of Algorithm, 2.

It can be observed from figure 6 that the classification accuracy of ELI augmented models generally increases as the clustering accuracy of algorithm 2 increases. For example, one can observe that the classification accuracy of SGC-ELI and LP-ELI on Cora (figure 6(a)) is higher than their classification accuracy on Wiki (figure 6(d)) just as the clustering accuracy of algorithm 2 is higher on Cora than on Wiki. **We want to emphatically note here that the clustering accuracy is not the same as the classification accuracy, as the classes returned by the clustering model may be a shuffled version of the true classes. For example, even on Cora where algorithm 2 achieves around 69% clustering accuracy, it has below 20% classification accuracy**

Figure 7 shows experiment results for the setting where the number of labeled nodes used for training are selected on a per-class basis i.e., $l_R$ for ELI is still selected as in Section 4.2, while $l = L_R + l_H$ is selected on a per-class basis for non-ELI augmented models. It can Equally be observed from figure 7 that even in this setting, the classification accuracy of ELI augmented models generally outperforms that of non-ELI augmented models by about 10-20% on most datasets when the number of labeled nodes used for training are extremely few. Moreover, it can also be observed that in this setting too, the performance of ELI-augmented models increases as the clustering accuracy of algorithm 2 increases.

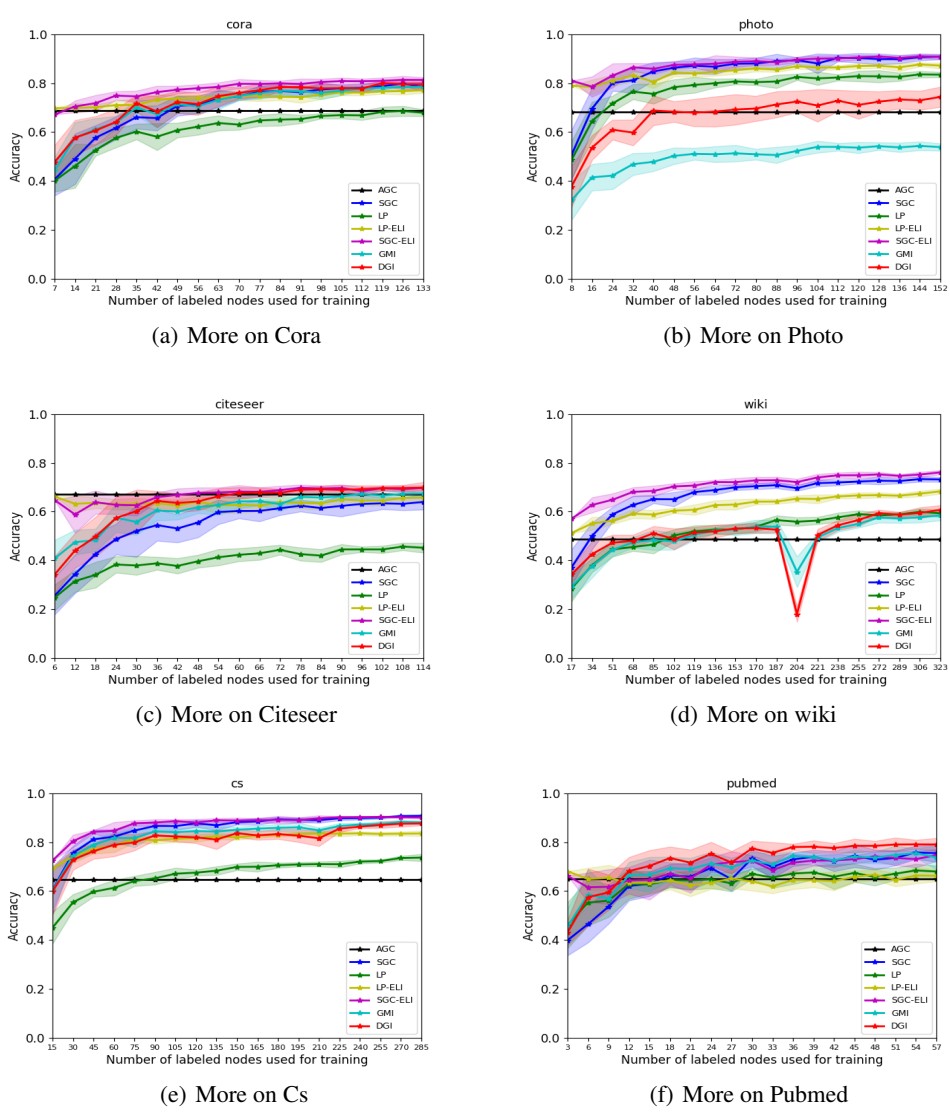

(a) More on Cora

(b) More on Photo

(c) More on Citeseer

(d) More on wiki

(e) More on Cs

(f) More on Pubmed

Figure 6: More extensive SLNC experiments on Cora, Pubmed, Citeseer, Photo, Wiki, and Cs (Computers not shown due to space, Computers similar to Citeseer). See parameter settings in Sections 5.5.

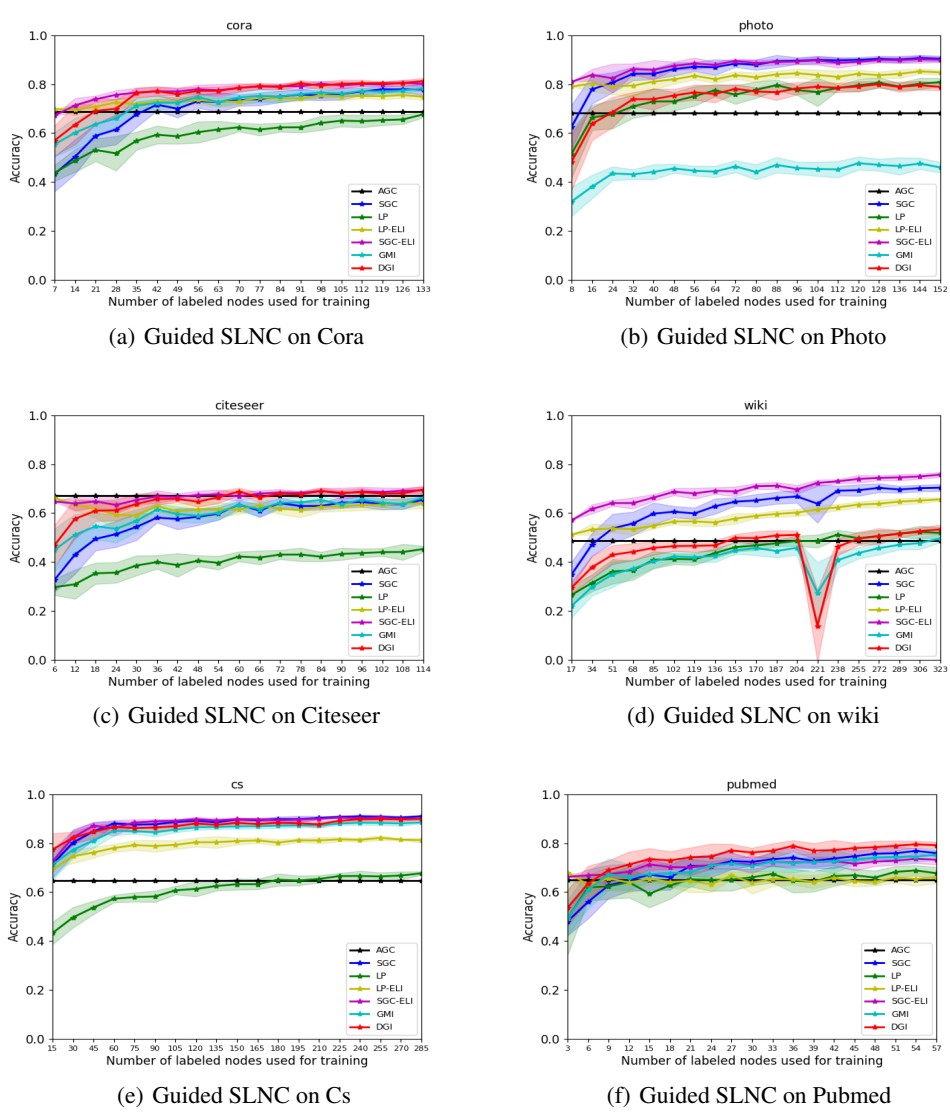

Figure 7: SLNC with $l_R$ (in Section 4.2) selected on a per-cluster basis for ELI, and $l = L_R + l_H$ selected on a per-cluster basis for non-ELI augmented models (Computers not shown due to space, Computers similar to Citeseer). See parameter settings in Sections 5.5.

