# OpenReview forum: "Sparse Labels Node Classification: Unsupervised Learning for Mentoring Supervised Learning in Sparse Label Settings"
_ICLR.cc/2024/Conference — Submitted to ICLR 2024_

### Official Review · Reviewer_g7gp · 2023-10-26

**Soundness:** 2 fair
**Presentation:** 1 poor
**Contribution:** 2 fair
**Rating:** 3
**Confidence:** 4

**Summary:**

This paper focuses on the problem of Sparse Label Node Classification (SLNC). Specifically, the authors first introduce a framework for estimating the label distribution information of nodes on the entire graph, and then select representative nodes for pseudo-labeling. The proposed method is integrated into existing models (SGC, LP) and experiments are conducted on seven datasets to validate its effectiveness.

**Strengths:**

1. The paper proposes a challenging graph semi-supervised learning scenario, where labels are sparser and more random (some classes may not even have labels at training time).
2. The paper provides relevant code and has a certain degree of credibility.
3. The paper conducts relevant experiments on 7 public benchmark datasets to verify its effectiveness.

**Weaknesses:**

1. The writing is not standardized, with too many formatting, grammatical, and presentation errors. For example, the tenses in the introduction part of the experiment are not uniform, and symbols are repeatedly used.
2. The solutions to the given field problem are: 1) Capture the label distribution information in the graph, and 2) Pseudo-label the representative nodes. For the former, the paper does not give an explanation based on intuition or related theory but only introduces existing strategies. For the latter, this is a common method in semi-supervised scenarios. At the same time, the introduction to the selection of representative nodes is not clear.
3. The paper proposes a semi-supervised learning scenario under sparse labels, but the experimental part is not clear about the setting of this scenario.
4. As far as I know, there are related works that consider graph learning under limited labels, such as CGPN[1], but this article does not mention and compare
5. In the experiment, the comparison methods are not new enough. The display of experimental results is not intuitive enough, there is no specific data, only charts. Meanwhile, the analysis of the experiment is unclear and there are many complex model names, such as in 5.7. It is recommended to simplify it.


[1] Wan, Sheng, et al. "Contrastive graph poisson networks: Semi-supervised learning with extremely limited labels." Advances in Neural Information Processing Systems 34 (2021): 6316-6327.

**Questions:**

1. The paper proposes that the label distribution information in the graph needs to be captured to assist classification. What is the intuition behind this approach? A new filter is introduced in the approach, which can make the large eigenvalues in the Laplacian matrix smaller. Why do we do this? At the same time, how is the parameter k set?
2. The paper proposes that pseudo-labels should be given to representative nodes. Where is the representativeness reflected here?
3. Why is the introduction of related work not divided into chapters? How relevant is it to your question?
4. Regarding the design of LA in Chapter 4.3, it is necessary to maintain at least three N*N adjacency matrices. Will this bring a large amount of calculation and memory usage? Please analyze the complexity of the algorithm.
5. The paper gives a set of experimental hyper-parameters in 5.5.1, such as learning rate, hidden layer size, etc. Does this mean that the settings are the same for each data set?
6. It is recommended that the author compares the method with semi-supervised learning models and advanced methods. Meanwhile, the experimental results need to be clearly presented.

---

> ### Author Response · Authors · 2023-11-10
> **response to Reviewer g7gp**
>
> Dear Reviewer, we thank you for your diligence in reading our work and giving us feedback. We hope to convince you of the pertinence and importance of our work through our response. Please find our preliminary responses to your concerns (please should you have more, do not hesitate to ask):
>
> W1) The writing is not standardized, with too many formatting, grammatical, and presentation errors. For example, the tenses in the introduction part of the experiment are not uniform, and symbols are repeatedly used.
> - We thank the reviewer for this concern. We sincerely apologize for this and will address all the formatting issues in the revision.
>
> W2) The solutions to the given field problem are: 1) Capture the label distribution information in the graph, and 2) Pseudo-label the representative nodes. Fora the former, the paper does not give an explanation based on intuition or related theory but only introduces existing strategies. For the latter, this is a common method in semi-supervised scenarios. At the same time, the introduction to the selection of representative nodes is not clear.
> - We apologize that in the current version the motivation for 1) ELI is not clear enough (as other reviewers pointed out as well).  ELI is proposed due to the points below:
> - - (a) In the real world, there exist settings in which it is prohibitive to have more labeled nodes (consider the Facebook graph for example, or the YouTube graph), and in such scenarios, one will be in a sparse labeled node classification (SLNC) setting.
> - - (b) We observe from the demo datasets we used that existing GNNs (given the baselines we used) cannot handle such a scenario. Since (1) in such a scenario, randomly selected nodes whether on a per-class basis or randomly over all classes may not be representative of the classes in the graph, and (b) The message-passing framework of GNNs may not be well-adapted given that the label distribution over the entire graph may not be well captured by the adjacency matrix of the graph.
> - - (c) As such one will need to carefully select key nodes as well as estimate the label distribution of the graph to hope to perform well.
> - - (d) Following this we propose ELI. We show in the appendix that the critical component in ELI is not the key labeled nodes but the label distribution over the graph (see the Ablation in figure 4 in section 5.7 that pink plots which have no Key labeled nodes (no KL) but have the KNN graph (KG) perform better than the yellow plots which have the KL but no KG). We will try to improve this in the paper revision during the coming week
> - To select the key labeled nodes, we select $l_H$ nodes that have the smallest clustering loss (i.e., the nodes that capture best the pseudo label classes, we select 1 for each pseudo-class where a pseudo-class in our work is a cluster).
>
> W3) The paper proposes a semi-supervised learning scenario under sparse labels, but the experimental part is not clear about the setting of this scenario.
> - We apologize if the experiment section is not clear enough, we will clarify this in the revision. From the x-axis of the plots in Figures 1, 3, 4, 6, and 7 one can see that the number of labeled nodes used for training increases gradually, and we plot this against the models' accuracy to show the effectiveness of ELI when only very few labeled nodes are used at training time. Moreover, we conduct ablation studies on ELI to prove that ELI as a whole (i.e., with the label distribution (KG) and with the key labels (KL)) performs best, while the model with the KG only performs second best, thus showing the importance of the label distribution in SLNC setting.
>
> W4) As far as I know, there are related works that consider graph learning under limited labels, such as CGPN[1], but this article does not mention and compare:
> - We sincerely apologize that we were not aware of this work or others for the SLNC setting.
> - At first glance, our intuition on this work is that:
> - - (a) Indeed it is a great work and we will include it in our main paper in citation and some comparisons.
> - - (b) it is interesting to see that they too try to estimate a posteriori distribution of labels just like we did in our ELI framework, which we are happy to know as it comforts our idea and approach as well
> - - (c) when nodes are selected randomly on a per-class basis, from the results in the paper both on Citeseer and Cora for 1 label per class to around 4 labels per class (which are shown in the paper), our simple SGC-ELI will perform better than it but we will post more accurate results here and in the paper after experiments (as this is a preliminary response). Our observation is that

---

> ### Author Response · Authors · 2023-11-10
> **More responses to Reviewer g7gp**
>
> W5) In the experiment, the comparison methods are not new enough. The display of experimental results is not intuitive enough, there is no specific data, only charts. Meanwhile, the analysis of the experiment is unclear and there are many complex model names, such as in 5.7. It is recommended to simplify it.
> - We will definitely modify section 5.7 and perhaps move it to the appendix for more space needed for clarity. We used charts due to the sheer number of experiments run. We will add tables to the appendix as suggested.
>
> Q1) The paper proposes that the label distribution information in the graph needs to be captured to assist classification. What is the intuition behind this approach? A new filter is introduced in the approach, which can make the large eigenvalues in the Laplacian matrix smaller. Why do we do this? At the same time, how is the parameter k set?
> - We apologize that in the current version the motivation for 1) ELI is not clear enough (as other reviewers pointed out as well). ELI is proposed due to the points below:
> - -(a) In the real world, there exist settings in which it is prohibitive to have more labeled nodes (consider the Facebook graph for example, or the YouTube graph), and in such scenarios, one will be in a sparse labeled node classification (SLNC) setting.
> - - (b) We observe from the demo datasets we used that existing GNNs (given the baselines we used) cannot handle such a scenario. Since (1) in such a scenario, randomly selected nodes whether on a per-class basis or randomly over all classes may not be representative of the classes in the graph, and (b) The message-passing framework of GNNs may not be well-adapted given that the label distribution over the entire graph may not be well captured by the adjacency matrix of the graph.
> - - (c) As such one will need to carefully select key nodes as well as estimate the label distribution of the graph to hope to perform well.
> - - (d) Following this we propose ELI. We show in the appendix that the critical component in ELI is not the key labeled nodes but the label distribution over the graph (see the Ablation in figure 4 in section 5.7 that pink plots which have no Key labeled nodes (no KL) but have the KNN graph (KG) perform better than the yellow plots which have the KL but no KG). We will try to improve this in the paper revision during the coming week
> - To select the key-labeled nodes, we select $l_H$ nodes that have the smallest clustering loss (i.e., the nodes that capture best the pseudo-label classes, we select 1 for each pseudo-class where a pseudo-class in our work is a cluster).
> - Once the Label distribution is obtained, we use it to build a new composite laplacian from three views (the original graph adjacency view, the label distribution adjacency view, and the view of the connectivity ground truth training labeled), we then use this new laplacian to build a new lowpass filter as has been proven to be useful for feature denoising\message propagation\label propagation in GNNs and use this lowpass filer as to aggregate our features before feeding them to a neural network or using a traditional heuristic such as label propagation.
>
> Q2) The paper proposes that pseudo-labels should be given to representative nodes. Where is the representativeness reflected here?
> - In our ELI framework,  (1) we estimate the label distribution over the entire graph by estimating pseudo labels for each node (where in our current framework pseudo labels are cluster assignments), and then use this distribution to build the composite laplacian, and we use this label distribution to get representative node where the representative nodes are nodes that represent each pseudo-lass (cluster well), for this we selected the nod per cluster with the smallest distance from the centroid as explained in section 4.2
>
> Q3) Why is the introduction of related work not divided into chapters? How relevant is it to your question?
> - We apologize that we do not understand the concern well, from our understanding, the reviewer is asking what the main point of our related work section is and as such why is it organized the way it is?
> - - if yes, then the main point of our related work section is to introduce related works from different GNN domains, as well as discuss why they are not suited to the SLNC setting. We will read this section again since it seems the reviewer is suggesting that it may not be very clear.

---

> ### Author Response · Authors · 2023-11-10
> **More responses to Reviewer g7gp**
>
> Q4) Regarding the design of LA in Chapter 4.3, it is necessary to maintain at least three N*N adjacency matrices. Will this bring a large amount of calculation and memory usage? Please analyze the complexity of the algorithm.
> - No we do not need to keep all three of them, but the weighted laplacian as discussed in section  4.4
> - the time complexity of the algorithm (both 1 and 2, noting that 2 can be precomputed) is as in (a) and (b) below
> - - (a) the total time complexity for LP-ELI is $O(cN + cnd + knd + dn^2 + tcnd)$, where $cN$ is for equation (5), $O(nd + kn)$ is for the KNN graph, $O(dn^2)$ is for the SVD, and $O(tcnd) = O(nd)$ is for kmeans since t and c are extemely small
> - - (b) the total time complexity for SGC-ELI is  $O(Nd + ndc + cnd + knd + dn^2 + tcnd)$, where Nd if for the filtering operation $\hat{\mathbf{X}} = \mathcal{N}(\mathbf{L}_{A})\mathbf{X}$ defined in section~4.5, ndc for multipling the filtered features and the neurons of size d by c i.e, $\hat{\mathbf{X}}\mathbf{W}$,  and the rest being similar to those defined for the LP-ELI model. Giving a simplified complexity of $O(dn^2 + ndc +knd)$, with $c<<d<<N << n^2$ and $t$ being a constant. We will add this to the appendix as well
>
> Q5) The paper gives a set of experimental hyper-parameters in 5.5.1, such as learning rate, hidden layer size, etc. Does this mean that the settings are the same for each data set?
> - Yes the hyperparameters of ELI-enhanced models and all models in section 5.5.1 are fixed for all datasets as we aimed to make ELI as hyperparameter-free as possible
>
> Q6) It is recommended that the author compares the method with semi-supervised learning models and advanced methods. Meanwhile, the experimental results need to be clearly presented.
> - We will definitely add at least two newer models in the updated version.
> - We will add tables for the shorter experiments from 1-4 labels per class (random and random-per-class) to the appendix by next Tuesday Eve or Wednesday Eve.

---

> ### Author Response · Authors · 2023-11-20
> **Additional Preliminary Response to Reviewer g7gp**
>
> We are currently comparing with :
> - Wan, Sheng, et al. "Contrastive graph poisson networks: Semi-supervised learning with extremely limited labels." Advances in Neural Information Processing Systems 34 (2021): 6316-6327.
> - We noticed a huge performance gap between our result on their CGPN result and the ones reported in their paper. So we checked their paper again and noticed actually their setting requires the nodes to be selected on a per-class basis. As such, our ELI and even other baselines will outperform them when this is not the case. We will release the updated paper tomorrow with new results. Additionally, their model takes a huge amount of time to run. We will discuss these in the main paper as well as it is a closely related work.

---

> ### Author Response · Authors · 2023-11-22
> **Revised paper notification to Reviewer g7gp**
>
> The revised paper has been uploaded, the changes are in blue. Please let us know if you have any further questions

---

### Official Review · Reviewer_zMM3 · 2023-10-27

**Soundness:** 3 good
**Presentation:** 2 fair
**Contribution:** 2 fair
**Rating:** 5
**Confidence:** 4

**Summary:**

This paper proposes an effective graph-based semi-supervised learning approach for sparsely labeled nodes. The proposed approach uses an unsupervised learning approach to compute a pseudo-label distribution and a semi-supervised learning approach to estimate labels. The paper conducted experiments to show the effectiveness of the proposed approach.

**Strengths:**

- I like the paper's motivation; it is quite a fundamental research problem to improve the accuracy of graph neural networks.
- This paper is well structured and easy to follow.
- The related works are well presented in the paper.

**Weaknesses:**

- The proposed approach should be compared to more recent approaches.
- The parameter setting of the proposed approach is not justified in the paper.
- The theoretical properties of the proposed approach should be discussed in the paper.

**Questions:**

Why is the proposed approach effective for sparse label node classification? The motivation for the method described in the paper is unclear from the descriptions of the paper.

In the experiment, this paper compared the proposed approach to LP, SGC, DGI, GMI in which GMI is the most recent approach published in 2020. However, the compared approaches are not state-of-the-art. Since graph neural network is one of the popular research topics in machine learning, the proposed approach should be compared to more recent approaches. Could you compare the proposed approach to more recent approaches to show its effectiveness? The proposed approach should be compared to the approaches listed in Section 2.

As described in Section 4.4, beta_1, beta_2, and beta_3 are set to 1/3 due to the simplicity. However, it is unclear why this parameter setting is recommended in the paper. The proposed approach can handle other parameter settings. Why is this parameter setting recommended in the paper?

The paper should reveal the theoretical property of the proposed approach. As described in Section 4.3, it needs a high computation cost to compute Equation (4) directly. Therefore, the proposed approach approximately computes it using an iterative solution, as shown in Equation (5). I am interested in the theoretical difference between direct and approximate computations. Similarly, the proposed approach approximately computes the adjacency A_GH, as shown in Section 4.4. Please discuss the theoretical properties of the approximate computations.

In addition, I am interested in the theoretical computational cost and empirical processing time of the proposed approach. The proposed approach is more efficient than the previous approaches?

---

> ### Author Response · Authors · 2023-11-10
> **response to Reviewer zMM3**
>
> Dear Reviewer, we thank you for your diligence in reading our work and giving us feedback. We hope to convince you more on the pertinence of our work through our response. Please find our preliminary responses to your concerns (please should you have more, do not hesitate to ask):
>
> Q1) Why is the proposed approach effective for sparse label node classification? The motivation for the method described in the paper is unclear from the descriptions of the paper.
> - An intuitive motivation is provided in the introduction to section 4. However, we thank the reviewer for pointing out the fact that more explanation is indeed needed.
> - Our key observation is that in an extremely Sparse label setting, there may be a strong bias in the labels selected for training, as these labels may not be representative enough for their respective classes, as such, there may be a serious difference in the training domain and the testing domain (similar to the out of distribution setting), but in this case, dealing with extremely few labels available for training. As such we need first to estimate somewhat the way the labels are distributed over the entire graph (since most GNNs are message passing, so as to connect nodes with similar labels), as well as to select representative nodes for each of these classes. We will try rewrite this in the main paper.
>
> Q2) In the experiment, this paper compared the proposed approach to LP, SGC, DGI, GMI in which GMI is the most recent approach published in 2020. However, the compared approaches are not state-of-the-art. Since graph neural network is one of the popular research topics in machine learning, the proposed approach should be compared to more recent approaches. Could you compare the proposed approach to more recent approaches to show its effectiveness? The proposed approach should be compared to the approaches listed in Section 2.
> - We will include two newer models in the updated paper by Tuesday (should the reviewer have a specific model in mind, please do suggest here). Due to space, we will add this to the appendix.
>
> Q3) As described in Section 4.4, beta_1, beta_2, and beta_3 are set to 1/3 due to the simplicity. However, it is unclear why this parameter setting is recommended in the paper. The proposed approach can handle other parameter settings. Why is this parameter setting recommended in the paper?
> - There is no reason for this suggestion besides it being an assumption that we consider all three information sources of equal value given that we were attempting to make the model less cumbersome (in terms of number of hyperparameters). In fact if one is confident enough of the Label Distribution Estimation model (i.e., the model from which the KNN graph is obtained), one may increase the weight for its normalized adjacency \hat{\mathbf{A}}_{\mathcal{G}_H}, as well as increase the weight for the normalized ground truth label adjacency  \hat{\mathbf{A}}_{\mathcal{G}_Y} as the number of training labels increase (since in this case the label adjacency indeed will correspond more and more to the ground truth distribution of labels). We will  include this to the appendix as well
>
> Q4) The paper should reveal the theoretical property of the proposed approach. As described in Section 4.3, it needs a high computation cost to compute Equation (4) directly. Therefore, the proposed approach approximately computes it using an iterative solution, as shown in Equation (5). I am interested in the theoretical difference between direct and approximate computations. Similarly, the proposed approach approximately computes the adjacency A_GH, as shown in Section 4.4. Please discuss the theoretical properties of the approximate computations.
> - First, for the close-form solution in (4)  vs the approximation in (5):
> - - (a) the closed-form in (4) needs $O(n^3)$ for the matrix inverse and an additional $O(cn^2)$ where c is the number of classes, giving a total of $O(n^3 + cn^2)$ time complexity,
> - - (b) assuming that \hat{\mathbf{A}}_{\mathbf{A}} has $N$ nonzero elements, the closed-form in (5) has a complexity of $O(tNc)$ where $t$ is the number of iterations and $c$ the number of classes ($t$ was set to $60 << n$) in the paper.
> - Second computing \hat{\mathbf{A}}_{\mathcal{G}_H} from \mathbf{HH^{T}} has a complexity of $O(cn^2)$ if \mathbf{H} is dense using while using the KNN graph built from the soft cluster matrix \mathbf{U} as an approximation to \mathbf{HH^{T}} has a complexity of $O(nd + kn)$, where $k$ is the number of neighbors, and $d$ the number of node features, and in this case, the matrix \hat{\mathbf{A}}_{\mathcal{G}_H}, will be sparse since we can control the number of nodes to connect as shown in figure 2, and in addition since we are not using hard clusters for the graph, there will be some edges between close clusters. We will include these to the appendix as well

---

> ### Author Response · Authors · 2023-11-10
> **response to question 5 of Reviewer zMM3**
>
> Q5) In addition, I am interested in the theoretical computational cost and empirical processing time of the proposed approach. The proposed approach is more efficient than the previous approaches?
> - the total time complexity for LP-ELI is $O(cN + cnd + knd + dn^2 + tcnd)$, where $cN$ is for equation (5), $O(nd + kn)$ is for the KNN graph, $O(dn^2)$ is for the SVD, and $O(tcnd) = O(nd)$ is for kmeans since t and c are extemely small
> - the total time complexity for SGC-ELI is  $O(Nd + ndc + cnd + knd + dn^2 + tcnd)$, where Nd if for the filtering operation $\hat{\mathbf{X}} = \mathcal{N}(\mathbf{L}_{A})\mathbf{X}$ defined in section~4.5, ndc for multipling the filtered features and the neurons of size d by c i.e, $\hat{\mathbf{X}}\mathbf{W}$,  and the rest being similar to those defined for the LP-ELI model. Giving a simplified complexity of $O(dn^2 + ndc +knd)$, with $c<<d<<N << n^2$ and $t$ being a constant. We will add this to the appendix as well

---

> ### Author Response · Authors · 2023-11-22
> **Revised paper notification to Reviewer zMM3**
>
> The revised paper has been uploaded, the changes are in blue. Please let us know if you have any further questions

---

### Official Review · Reviewer_L6C4 · 2023-10-31

**Soundness:** 2 fair
**Presentation:** 2 fair
**Contribution:** 1 poor
**Rating:** 3
**Confidence:** 4

**Summary:**

This paper studies the problem of node classification when there is a very limited number of labelled nodes. The authors propose to utilize existing unsupervised method for clustering attributed graphs as a heuristic to estimate the node labels, and then incorporate the estimated node labels into a label propagation procedure in the form of a regularizer in the optimization objective. The authors conduct experiments over several real-world datasets to demonstrate the effectiveness of their method.

**Strengths:**

- The problem of node classification in the sparse label regime is an interesting problem and can potentially have many practical applications.

**Weaknesses:**

- Overall, the proposed method is just a very heuristic combination of an existing unsupervised method for clustering attributed graphs and label propagation. Therefore, the novelty of this work is very limited. Moreover, I think that the performance of the proposed method will be heavily affected by the performance of the label estimation step. There is no principled guarantees for the proposed method to work well in general, especially when there are more labels than what the authors considered in the experiments. More on this in the next point.

- I am not really convinced by the empirical section that the sparse labels setting is important. (The sparse labels setting may be a relevant and important problem for many practical settings, but the empirical section fails to demonstrate that. ) The datasets used in the experiments are standard benchmarks for node classification on graphs. In the standard train/test splits, these datasets already have a very low label rate, ranging from ~1.5% on Computers to ~5% on Cora and CiteSeer. Existing methods already perform very well at this label rate. The authors considered scenarios where there are only 3 to 60 labels in these datasets. This looks like a made-up setting to me. I would recommend the authors find alternative datasets where it is genuinely hard to obtain many ground-truth node labels, and thus we can only have access to a few dozen labels.

- The clarity and writing can be significantly improved. When I read the paper, I often have to read a couple of places twice or three times in order to understand what the authors trying to explain. Let me just give an example. In Section 4.2, the authors wrote "..., nodes in $l_H$ were chosen ...". This is confusing due to previously $l_H$ was defined to be a number, but in this sentence $l_H$ seems to be a set of nodes. I would recommend the authors spend serious effort to polish the writing of this paper.

**Questions:**

- Based on the description given in Section 4.4, in your experiments, did you simply use $\mathbf{A}_{\mathcal{G}_H}$ as the adjacency matrix of the KNN graph, as opposed to $HH^T$? In that case, is the information in $H$ being used anywhere?

---

> ### Author Response · Authors · 2023-11-10
> **Response to Reviewer L6C4**
>
> Dear Reviewer, we thank you for your diligence in reading our work and giving us feedback. We hope to convince you of the pertinence and importance of our work through our response. Please find our preliminary responses to your concerns (please should you have more, do not hesitate to ask):
>
> $Q1)$ Overall, the proposed method is just a very heuristic combination of an existing unsupervised method for clustering attributed graphs and label propagation. Therefore, the novelty of this work is very limited. Moreover, I think that the performance of the proposed method will be heavily affected by the performance of the label estimation step. There is no principled guarantees for the proposed method to work well in general, especially when there are more labels than what the authors considered in the experiments. More on this in the next point.
> - We thank the reviewer for the concern on the motivation of our work. We indeed will clarify this in the revision. As the reviewer seems concerned about our main contributions, we find it necessary to restate them here and elaborate on them:
> - - (a) In the real world, there indeed exists extremely large graphs on which it is prohibitive to have more labeled nodes (consider the Facebook graph for example, or the YouTube graph), and in such scenarios, one will be in a sparse labeled node classification setting.
> - - (b) We observe from the demo datasets we used that existing GNNs (given the baselines we used) cannot handle such a scenario. Since (1) in such a scenario, randomly selected nodes whether on a per-class basis or randomly over all classes may not be representative of the classes in the graph, and (b) The message-passing framework of GNNs may not be well-adapted given that the label distribution over the entire graph may not be well captured by the adjacency matrix of the graph.
> - - (c) As such one will need to carefully select key nodes as well as estimate the label distribution of the graph to hope to perform well.
> - - (d) Following this we propose ELI. We show in the appendix that the critical component in ELI is not the key labeled nodes but the label distribution over the graph (see the Ablation in figure 4 in section 5.7 that pink plots which have no Key labeled nodes (no KL) but have the KNN graph (KG) perform better than the yellow plots which have the KL but no KG). We will try to move the ablation to the appendix for a clearer and more extensive discussion of this by Tuesday evening.
>
> $Q2)$ I am not really convinced by the empirical section that the sparse labels setting is important. (The sparse labels setting may be a relevant and important problem for many practical settings, but the empirical section fails to demonstrate that. ) The datasets used in the experiments are standard benchmarks for node classification on graphs. In the standard train/test splits, these datasets already have a very low label rate, ranging from ~1.5% on Computers to ~5% on Cora and CiteSeer. Existing methods already perform very well at this label rate. The authors considered scenarios where there are only 3 to 60 labels in these datasets. This looks like a made-up setting to me. I would recommend the authors find alternative datasets where it is genuinely hard to obtain many ground-truth node labels, and thus we can only have access to a few dozen labels.
> - Indeed we agree with the reviewer that some GNNs perform well with label rates of ~1.5% on Computers to ~5% on Cora and CiteSeer. However, our point was to prove that when the label rate is even lower, i.e.,  the SLNC setting, most existing GNNs will not perform well as shown on the same datasets on which they performed well in the semi-supervised setting. However, as the reviewer wisely suggested for completeness we will some larger graphs to demonstrate this fact. Please if the reviewer can suggest one or two such graphs it will be well appreciated.
>
> Q3) The clarity and writing can be significantly improved. When I read the paper, I often have to read a couple of places twice or three times in order to understand what the authors trying to explain. Let me just give an example. In Section 4.2, the authors wrote "..., nodes in were chosen ...". This is confusing due to previously was defined to be a number, but in this sentence seems to be a set of nodes. I would recommend the authors spend serious effort to polish the writing of this paper.
> - We sincerely apologize for the typos and clarity issues, we will fix these in the revision

---

> ### Author Response · Authors · 2023-11-10
> **response to question 4 of Reviewer L6C4**
>
> Q4) Based on the description given in Section 4.4, in your experiments, did you simply use \hat{\mathbf{A}}{\mathcal{G}_H}as the adjacency matrix of the KNN graph, as opposed to \mathbf{HH^{T}}? In that case, is the information in  \mathbf{H} being used anywhere?
> - Yes we simply used \hat{\mathbf{A}}{\mathcal{G}_H} rather \mathbf{HH^{T}}, as this will be sparse as we can control the number of nodes to connect  (figure 2).  Also, since we are not using hard clusters for the graph, there will be some edges between close clusters.

---

> ### Author Response · Authors · 2023-11-22
> **Revised paper notification to Reviewer L6C4**
>
> The revised paper has been uploaded,  the changes are in blue. Please let us know if you have any further questions

---

### Official Review · Reviewer_N5b9 · 2023-11-05

**Soundness:** 2 fair
**Presentation:** 2 fair
**Contribution:** 2 fair
**Rating:** 5
**Confidence:** 4

**Summary:**

This paper introduce a new task Sparse Labels Node Classification (SLNC) in graph learning. Compared to the existing task Semi-Supervised Node Classification (SSNC), only an extremely small portion of the labels are known, and the labels are not distributed equally across classes. The authors propose a new framework by estimating label information. The authors conduct experiments and show their proposed framework achieves performance better than existing approaches.

Post-rebuttal: I have read the rebuttal and would like to keep my scores.

**Strengths:**

The authors introduce a new task Sparse Labels Node Classification (SLNC) in graph learning and propose an Estimate Label Information (ELI) framework to solve this new task. Empirical results show the efficacy of the proposed approach.

**Weaknesses:**

While the authors claim that only an extreme small amount of labels are known in the newly proposed setting, it seems that at least a small portion of labels are actively selected---e.g., the first l_H data points are labeled according to the approach proposed in Section 4.2. What would happen if all labels are randomly selected?

The presentations of the paper can be improved. For instance, Algorithm 1 is not written in a clear way---one has to going back and forth between Algorithm 1 and Sections 4.2, 4.3, 4.4.

**Questions:**

See comments above.

---

> ### Author Response · Authors · 2023-11-10
> **response to Reviewer N5b9**
>
> Dear Reviewer, we thank you for your diligence in reading our work and giving us feedback. We hope to convince you more on the pertinence of our work through our response. Please find our preliminary responses to your concerns (please should you have more, do not hesitate to ask):
>
> Q1) While the authors claim that only an extremely small amount of labels are known in the newly proposed setting, it seems that at least a small portion of labels are actively selected---e.g., the first l_H data points are labeled according to the approach proposed in Section 4.2. What would happen if all labels were randomly selected:
> - We would love to first note that the support set l_H chosen is automatically chosen based on the clustering loss (i.e., the points closest to the cluster centroids, and as such is whole with the ELI framework, we call this the KL step (Key label selection step).
> - (a) Given this observation, in the Ablation in ~Section 5.7, we show that removing the KNN graph (i.e., the label distribution) represented by the yellow "no KG" plots has the worst effect, validating the need for this information for SLNC
> - (b) We equally show in Ablation in ~Section 5.7, the KL step (i.e., the estimation of l_H datapoints) represented by the pink "no KL" plots has a worse performance than full ELI (the green plots), while still outperforming the base model by itself (the blue plots)
> - points (a) and (b) show that both the KL and KG steps are indeed valuable in the SLNC setting
>
> Q2) The presentation of the paper can be improved. For instance, Algorithm 1 is not written in a clear way---one has to going back and forth between Algorithm 1 and Sections 4.2, 4.3, 4.4.
> - We apologize for the presentation issues, all these will be fixed in the updated paper by Tuesday eve.

---

> ### Author Response · Authors · 2023-11-22
> **Revised paper notification to Reviewer N5b9**
>
> The revised paper has been uploaded,  the changes are in blue. Please let us know if you have any further questions

---

### Meta-Review · Area_Chair_Loe7 · 2023-12-09

**Metareview:**

I recommend to reject this paper.

In this paper, the authors focused on the Sparse Labels Node Classification (SLNC) in graph learning, where the number of lab  led nodes is very limited and proposed Estimate Label Information (ELI) for this problem leveraging existing unsupervised clustering methods on the attribute graph to estimate node labels and label propagation techniques with the estimated node labels.

After the rebuttal period, all the reviewers still think that this paper is below the acceptance threshold due to the weakness: 1) lack of enough comparison with more recent GNN approaches; 2) writing and presentation is not standardized; 3) limited novelty of the work without enough theoretical analysis/insights. As a result, I would recommend to reject this paper and encourage the authors to take the reviewers' feedbacks to further revise the work.

**Justification For Why Not Higher Score:**

N/A

**Justification For Why Not Lower Score:**

N/A

---

### Decision · Program_Chairs · 2024-01-16

Reject